# Sheaves Reloaded: A Directional Awakening

**Stefano Fiorini**[1][*][†]    **Hakan Aktas**[2][*]    **Iulia Duta**[2]    **Pietro Morerio** [1]

**Alessio Del Bue**[1]    **Pietro Lio**[2]    **Stefano Coniglio**[4]

[1] Istituto Italiano di Tecnologia, Genoa, Italy
[2] University of Cambridge, Cambridge, United Kingdom
[3] University of Bergamo, Bergamo, Italy

## Abstract

Sheaf Neural Networks (SNNs) are a powerful algebraic-topology generalization of Graph Neural Networks (GNNs), and have been shown to significantly improve our ability to model complex relational data. While the GNN literature proved that incorporating directionality can substantially boost performance in many real-world applications, no SNN approaches are known with such a capability. To address this limitation, we introduce the Directed Cellular Sheaf, a generalized cellular sheaf designed to explicitly account for edge orientations. Building on it, we define a corresponding sheaf Laplacian, the Directed Sheaf Laplacian $L^{\widetilde{\mathcal{F}}}$, which exploits the sheaf's structure to capture both the graph's topology and its directions. $L^{\widetilde{\mathcal{F}}}$ serves as the backbone of the Directed Sheaf Neural Network (DSNN), the first SNN model to embed a directional bias into its architecture. Extensive experiments show that DSNN consistently outperforms many baseline methods. The source code can be found at *https://github.com/hakanaktas0/DSNN*.

## 1 Introduction

The rapid advances in neural networks and deep learning have provided powerful tools for capturing complex relationships in structured data. Rooted in algebraic topology, Sheaf Neural Networks (SNNs) (Hansen & Ghrist, 2019; Bodnar et al., 2022) have recently emerged as a principled extension of traditional Graph Neural Networks (GNNs). They leverage the notion of a *cellular sheaf*, which equips a graph with vector spaces associated to its nodes and edges, together with *restriction maps* that relate the two when they are incident. This framework enables learning in higher-dimensional feature spaces, naturally mitigates oversmoothing, and improves performance in heterophilic graphs (where neighboring nodes may carry dissimilar features) (Bodnar et al., 2022).

Despite their strength, the SNNs proposed so far are limited to undirected graphs and overlook edge orientations, therefore failing to fully capture the graph topology of many real-world (naturally directed) applications. Indeed, directionality plays a central role in complex networks (Bianconi et al., 2008), underpinning topological and dynamical phenomena that can strongly influence system behavior (Harush & Barzel, 2017; Asllani et al., 2018). To amend this, we introduce a principled extension of SNNs which explicitly incorporates edge directionality. We do so by designing complex-valued direction-aware restriction maps and a corresponding directed sheaf Laplacian. With this, our method brings the expressive power of SNNs to the domain of directed graphs, enabling a richer message passing which respects any asymmetries in the graph relationships. By explicitly modeling edge orientation, our framework combines the benefits of SNNs' higher-dimension learning and robustness to heterophily with the advantages offered by directed GNNs (Zhang et al., 2021b). This makes our approach both theoretically principled and practically impactful for a wide range of applications, from social and biological networks to causal and flow-based systems.

We enhance the representational power of SNNs in settings where edge directionality is crucial by introducing the notion of *Directed Cellular Sheaves*. Unlike traditional cellular sheaves employed in

---

[*]Equal contribution.
[†]Corresponding author: `stefano.fiorini1994@gmail.com`

state-of-the-art SNNs, which assign vector spaces (or more general algebraic structures) to the cells of a complex without retaining a notion of direction, our framework incorporates direction explicitly into the sheaf's restriction maps. In it, we define the *Directed Coboundary Operator* $\tilde{\delta}$ associated with the Directed Cellular Sheaf, which we use to construct the *Directed Sheaf Laplacian* (DSL) operator $L^{\tilde{\mathcal{F}}}$, capturing both the graph's topological structure and the orientation of its edges.

**Our main contributions are the following ones:**

- We introduce the *Directed Cellular Sheaf*, a mathematical construct that enriches directed graphs by enabling a principled representation of directional interactions between its nodes. This structure assigns linear maps between the vector spaces associated with the graph's edges and vertices in such a way that the edge directions are explicitly represented.
- We propose the *Directed Sheaf Neural Network* (DSNN)—an SNN architecture explicitly designed to include an inductive bias that reflects the directional structure of the graph.
- We conduct extensive experiments on real-world and synthetic datasets, demonstrating the advantages of our proposal to incorporate directionality in an SNNs via the Directed Cellular Sheaf and its Laplacian operator $L^{\tilde{\mathcal{F}}}$.

## 2 BACKGROUND & RELATED WORK

### 2.1 CELLULAR SHEAVES

In the classical setting, a *sheaf* assigns data (such as sets, groups, or vector spaces) to open sets of a topological space (such as points, open segments, and open disks), together with restriction maps that propagate this data to open subsets within them. A *cellular sheaf* (Shepard, 1985; Curry, 2014) modifies this perspective by replacing open sets with cells of a cell complex (where 0-cells are points, 1-cells edges, 2-cells faces, etc.). It assigns a vector space to each cell and a linear restriction map from each higher-dimensional cell to each of its faces, reflecting the hierarchical structure of the complex. In line with recent works on SNNs (Hansen & Ghrist, 2019; Bodnar et al., 2022), we focus on cell complexes consisting only of 0-cells and 1-cells, which coincide with the nodes and edges of a graph, and on lower-to-higher dimensional mappings from nodes to edges. In such models, the sheaf structure enables a principled generalization of message-passing architectures by allowing node features to propagate through edge-level transformations governed by linear restriction maps.

Following Hansen & Ghrist (2019), we define the *cellular sheaf* of an undirected graph $G = (V, E)$ with $n = |V|$ and $m = |E|$ as the triple $(\{\mathcal{F}(u)\}_{u \in V}, \{\mathcal{F}(e)\}_{e \in E}, \{\mathcal{F}_{u \trianglelefteq e}\}_{e \in \Gamma(u)})$, containing a vector space $\mathcal{F}(u)$ associated with each vertex $u \in V$, a vector space $\mathcal{F}(e)$ associated with each edge $e \in E$, and a linear map $\mathcal{F}_{u \trianglelefteq e} : \mathcal{F}(u) \to \mathcal{F}(e)$ for each edge $e \in \Gamma(u)$, where $\Gamma(u)$ is the subset of edges incident on $u$. In line with the SNN literature, all vector spaces are assumed to be real. In the cellular sheaf, the vector spaces are referred to as *stalks*, while the linear maps are called *restriction maps*. In this framework, the vertex stalks $\mathcal{F}(u)$ represent the node feature vectors (traditionally denoted as $x_v$ in the graph-learning literature). The space formed by all the spaces associated with the nodes (resp., edges) of the graph is called the space of 0-cochains $C^0(G; \mathcal{F}) = \oplus_{u \in V} \mathcal{F}(u)$ (resp., the space of 1-cochains $C^1(G; \mathcal{F}) = \oplus_{e \in E} \mathcal{F}(e)$). The inter-vertex constraints are captured by the *coboundary operator* $\delta : C^0(G; \mathcal{F}) \to C^1(G; \mathcal{F})$, which, given an arbitrary orientation on the edges (where, for each $e = \{u, v\} \in E$, either $\mathcal{F}_{u \trianglelefteq e}$ or $\mathcal{F}_{v \trianglelefteq e}$ is multiplied by $-1$), is defined as $\delta(x)_e = \mathcal{F}_{u \trianglelefteq e} x_u - \mathcal{F}_{v \trianglelefteq e} x_v$. From the coboundary operator, one can define the *sheaf Laplacian* as $L^{\mathcal{F}} = \delta^T \delta$ which, for a given $x \in C^0(G; \mathcal{F})$, reads:

$$L^{\mathcal{F}}(x)_u = \sum_{e = \{u, v\}} \mathcal{F}_{u \trianglelefteq e}^T (\mathcal{F}_{u \trianglelefteq e} x_u - \mathcal{F}_{v \trianglelefteq e} x_v) \qquad \forall u \in V.$$

Both $L^{\mathcal{F}}$ and its normalized version $L_N^{\mathcal{F}}$ are positive semidefinite operators on the space of 0-cochains $C^0(G; \mathcal{F})$, and are independent of the chosen edge orientation, mirroring a similar property that holds for the standard graph Laplacian $L$ (Chung, 1997).

Several approaches have explored the use of sheaves in the context of graph-based learning. The first SNN was introduced by Hansen & Ghrist (2019), and later extended by Bodnar et al. (2022), who proposed the Neural Sheaf Diffusion (NSD) model. More recent SNN models build upon the NSD

framework, incorporating attention mechanisms (Barbero et al., 2022), extending the architecture to hypergraph data (Duta et al., 2023), and introducing nonlinearities (Zaghen et al., 2024).

The SNN literature assumes that all node and edge stalks are finite-dimensional vector spaces of dimension $d$, all of which are isomorphic to $\mathbb{R}^d$. In this way, every restriction map coincides with a $d \times d$ matrix. As a result, the sheaf Laplacian is a block-matrix of size $nd \times nd$ with blocks of size $d \times d$ which operates on an $nd$-dimensional vector-valued signal obtained by stacking the $d$-dimensional node signals $x_u \in \mathcal{F}(u)$ for all $u \in V$ associated with the graph's vertices (the 0-cochain). When considering multi-feature vertex signals with $f > 1$ features (or channels), a SNN operates on a matrix-valued graph signal of size $nd \times f$. For any $u, v \in V$, the block of indices $u, v$ of $L^F$ is equal to the $d \times d$ matrix $-\mathcal{F}^T_{u \trianglelefteq e} \mathcal{F}_{v \trianglelefteq e}$. The *sheaf Laplacian* generalizes the classical graph Laplacian on an undirected and unweighted graph $G$. This is because, in the special case of a *trivial sheaf*—a sheaf where each stalk is isomorphic to $\mathbb{R}$ and each restriction map is the identity map over $\mathbb{R}$—we recover the standard $n \times n$ graph Laplacian $L = D - A$, where $A \in \{0, 1\}^{n \times n}$ is the adjacency matrix, and $D := \mathrm{diag}(\mathbf{1}_n^\top A)$ where $\mathbf{1}_n$ is the all-one vector.

To the best of our knowledge, no SNNs, including those introduced in the above-mentioned papers, have been proposed to incorporate the edge directions directly. We set out to do so in this paper.

## 2.2 Discrete Laplacian matrices for undirected and directed graphs

In the literature, GNNs are typically classified into two categories: spectral-based and spatial-based (Wu et al., 2020). Spatial-based GNNs define the convolution as a localized-aggregation/message-passing operator (Wang et al., 2019). For example, GatedGCN (Li et al., 2016) handles directed graphs by aggregating information from out-neighbors (ignoring, though, potentially valuable signals from in-neighbors) and, more recently, Dir-GNN (Rossi et al., 2024) employs separate aggregation schemes with distinct weights for in-neighbors and out-neighbors. In contrast, spectral-based GNNs define the convolution operator rigorously as a function of the eigenvalue decomposition of the graph Laplacian (Kipf & Welling, 2017). Over the past few years, several approaches have been proposed to generalize spectral convolutions to directed graphs. In particular, DGCN (Tong et al., 2020b) introduces a first-order proximity matrix along with two second-order proximity matrices to describe both the neighborhood of each vertex and the vertices that are reachable from a given vertex in one hop. DiGCN (Tong et al., 2020a) adopts the Personalized PageRank matrix and incorporates $k$-hop diffusion matrices. Finally, several methods generalized the classical Laplacian matrix $L$ to suitably defined complex-valued, Hermitian matrices such as the Magnetic Laplacian (Lieb & Loss, 1993) and the Sign-Magnetic Laplacian (Fiorini et al., 2023).

The *Magnetic Laplacian* $L^{(q)}$, originally introduced by Lieb & Loss (1993) in the study of electro-magnetic fields and later employed in spectral GNNs by Zhang et al. (2021b;a), is a complex-valued Hermitian matrix that captures directional information in graphs while admitting an eigenvalue decomposition with a real, nonnegative spectrum. Letting $A_s := \frac{1}{2}\left(A + A^\top\right)$ be the symmetrized version of $A$ and letting $D_s := \mathrm{diag}(\mathbf{1}_n^\top A_s)$, the *Magnetic Laplacian* and its normalized version are defined as follows:

$$L^{(q)} := D_s - H^{(q)} \text{ and } L_N^{(q)} := I - D_s^{-\frac{1}{2}} H^{(q)} D_s^{-\frac{1}{2}}, \text{ with } H^{(q)} := A_s \odot \exp\left(\mathbf{i}\, 2\pi q\left(A - A^\top\right)\right),$$

where $\mathbf{i}$ is the imaginary unit and $q \in [0, 1]$.

The *Sign-Magnetic Laplacian* $L^\sigma$, introduced by Fiorini et al. (2023), is a Hermitian matrix that is well-defined even for graphs with negative edge weights and possesses several additional desirable properties. When $q = \frac{1}{4}$, $L^\sigma$ and $L^{(q)}$ coincide if the latter is first computed on the unweighted version of the graph and then element-wise multiplied by $A_s$. Thus, $L^\sigma$ is invariant to a positive weight scaling which could otherwise alter the sign pattern of $L^{(q)}$ and, thus, the edge direction. Letting $\bar{D}_s := \mathrm{diag}(\mathbf{1}_n^\top |A_s|)$ and $\mathrm{sign} : \mathbb{R}^{n \times n} \to \{-1, 0, 1\}^{n \times n}$ be the component-wise *signum* function, $L^\sigma$ and its normalized version are defined as follows:

$$L^\sigma := \bar{D}_s - H^\sigma \text{ and } L_N^\sigma := I - \bar{D}_s^{-\frac{1}{2}} H^\sigma \bar{D}_s^{-\frac{1}{2}}, \text{ with } H^\sigma := A_s \odot \left(e^\top - \mathrm{sgn}(|A - A^\top|) + \mathbf{i}\, \mathrm{sgn}\left(|A| - |A^\top|\right)\right).$$

## 3 Directed Cellular Sheaves and the Directed Sheaf Laplacian

In this paper, we introduce the notion of a *Directed Cellular Sheaf*, a special type of cellular sheaf where the node and edge stalks are vector spaces defined over the complex field and in which,

assuming finite-dimensional vector spaces, the restriction maps are either real-valued or complex-valued matrices where the latter encode the graph's direction. For clarity, we also include a notation table in Appendix A to help readers navigate the symbols we use throughout the paper.

## 3.1 DIRECTED CELLULAR SHEAF

For the ease of notation, we now introduce the Directed Cellular Sheaf for the case of finite-dimensional stalks (the definition can be easily extended to the infinite-dimensional case).

**Definition 1.** *The* Directed Cellular Sheaf *of a directed graph* $G = (V, E)$ *with adjacency matrix* $A \in \{0, 1\}^{n \times n}$ *is the tuple* $(T^{(q)}, \{\widetilde{\mathcal{F}}(u)\}_{u \in V}, \{\widetilde{\mathcal{F}}(e)\}_{e \in E}, \{\widetilde{\mathcal{F}}_{u \unlhd e}\}_{e \in \Gamma(u)})$ *consisting of:*

1. *A Hermitian matrix* $T^{(q)} := \exp(\mathbf{i} \, 2\pi q \left( A - A^\top \right))$, *parametric in* $q \in \mathbb{R}$.

2. *A vector space* $\widetilde{\mathcal{F}}(u) \in \mathbb{C}^d$ *associated with each vertex* $u \in V$;

3. *A vector space* $\widetilde{\mathcal{F}}(e) \in \mathbb{C}^d$ *associated with each edge* $e \in E$;

4. *Two linear maps* $\widetilde{\mathcal{F}}_{u \unlhd e}, \widetilde{\mathcal{F}}_{v \unlhd e}$ *that map* $\widetilde{\mathcal{F}}(u), \widetilde{\mathcal{F}}(v)$ *to* $\widetilde{\mathcal{F}}(e)$ *for each edge* $e \in E$ *with* $u \sim_e v$ *where* $\widetilde{\mathcal{F}}_{u \unlhd e} \in \mathbb{R}^{d \times d}$ *and* $\widetilde{\mathcal{F}}_{v \unlhd e} = \widetilde{\mathcal{F}}_{v \unlhd e}^0 T_{uv}^{(q)} \in \mathbb{C}^{d \times d}$, *with* $\widetilde{\mathcal{F}}_{v \unlhd e}^0 \in \mathbb{R}^{d \times d}$ *is a real-valued restriction map, and* $u \sim_e v$ *indicates that* $e$ *is incident to both* $u$ *and* $v$ *regardless of whether it is directed or not.*

An illustration is provided in Figure 1. The rationale of our definition is to encode the direction of each edge in the imaginary part of the restriction map of the tail node.

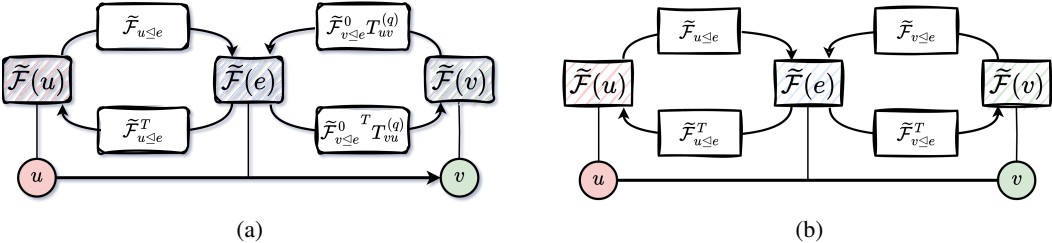

(a)                                                                 (b)

Figure 1: An illustration of the complex-valued restriction maps of the *Directed Cellular Sheaf* showing how they encode the graph's directionality for (a) a directed edge and (b) an undirected edge.

The core idea behind our Directed Sheaf Laplacian is that it is *complex Hermitian* operator whose *magnitude* captures undirected geometry while its *phase* encodes edge directions, with the parameter $q$ modulating the strength of this directional component. Maintaining a *PSD/Hermitian* structure is crucial for spectral GNNs: it guarantees real, non-negative eigenvalues, enabling stable Fourier bases and well-defined spectral filters. Classical spectral GNNs (e.g., GCN (Kipf & Welling, 2017)) also rely on PSD operators with bounded real spectra, but they cannot represent directionality. Our complex Hermitian formulation preserves the necessary PSD spectral properties *while introducing direction-aware phase information*, yielding a stable and expressive spectral operator for directed graphs. If we do not preserve the PSD/Hermitian structure, the spectral interpretation of graph convolutions breaks down. Specifically, a non-PSD matrix can have negative or complex eigenvalues, which makes the graph Fourier transform ill-defined for filtering purposes: the "frequencies" may no longer correspond to real oscillations on the graph, and spectral multipliers can produce unstable or non-convergent outputs. For instance, using a purely real skew-symmetric matrix (antisymmetric) yields purely imaginary eigenvalues, so applying a spectral filter results in oscillatory or diverging behavior rather than meaningful smoothing or directional propagation. For example, in the undirected case where $e = \{u, v\}$, we have $A_{uv} = A_{vu} = 1$, $T_{uv}^{(q)} = \cos(0) + \mathbf{i}\sin(0) = 1$ and, thus, $\widetilde{\mathcal{F}}_{v \unlhd e} = \widetilde{\mathcal{F}}_{v \unlhd e}^0$ are purely real. In the directed case where $e = (u, v)$ and assuming $q = \frac{1}{4}$, we have $A_{uv} = 1$ and $A_{vu} = 0$ and $T_{uv}^{(q)} = \cos(-\pi\frac{1}{2}) + \mathbf{i}\sin(-\pi\frac{1}{2}) = -\mathbf{i}$; thus $\widetilde{\mathcal{F}}_{v \unlhd e} = -\widetilde{\mathcal{F}}_{v \unlhd e}^0 \mathbf{i}$ and the sign of the imaginary part indicates the edge's direction. Our proposed Directed Cellular Sheaf generalizes the classical Cellular Sheaf since, if $G$ is undirected, $T_{vu}^{(q)} = 1$ for all $\{u, v\} \in E$ for any

choice of $q$ and, thus, the two sheaves coincide. If $G$ is directed, but we set $q = 0$, we obtain the classical Cellular Sheaf associated with the undirected version of $G$ (see Appendix J for an additional example).

Let $E^0 \cup E^1 = E$ be a partition of the edge set $E$ into undirected edges ($E^0$) and directed edges ($E^1$). We define the *Directed Coboundary Operator* $\widetilde{\delta}$ associated with the Directed Cellular Sheaf as $\widetilde{\delta}(x)_e := \widetilde{\mathcal{F}}_{u \trianglelefteq e} x_u - \widetilde{\mathcal{F}}_{v \trianglelefteq e} x_v, e \in E$, where $x$ is a cochain of the Directed Cellular Sheaf. Thanks to our definition of $\widetilde{\mathcal{F}}_{u \trianglelefteq e}, \widetilde{\mathcal{F}}_{v \trianglelefteq e}$, we have:

$$\widetilde{\delta}(x)_e = \begin{cases} \widetilde{\mathcal{F}}_{u \trianglelefteq e} x_u - \widetilde{\mathcal{F}}_{v \trianglelefteq e} x_v & \text{if } e \in E^0 \\ \widetilde{\mathcal{F}}_{u \trianglelefteq e} x_u - \widetilde{\mathcal{F}}^0_{v \trianglelefteq e} T^{(q)}_{uv} x_v & \text{if } e \in E^1. \end{cases} \tag{1}$$

We define the *Directed Sheaf Laplacian* (DSL) $L^{\widetilde{\mathcal{F}}}$ associated to a Directed Cellular Sheaf as $L^{\widetilde{\mathcal{F}}} := \widetilde{\delta}^* \widetilde{\delta}$, where $*$ is the conjugate transpose operator. Each $d \times d$ block of $L^{\widetilde{\mathcal{F}}}$ reads:

$$L^{\widetilde{\mathcal{F}}}_{uv} = \begin{cases} -\widetilde{\mathcal{F}}^*_{u \trianglelefteq e} \widetilde{\mathcal{F}}_{v \trianglelefteq e} = -\widetilde{\mathcal{F}}^T_{u \trianglelefteq e} \widetilde{\mathcal{F}}^0_{v \trianglelefteq e} T^{(q)}_{uv} & \text{if } e = (u, v) \\ -\widetilde{\mathcal{F}}^*_{u \trianglelefteq e} \widetilde{\mathcal{F}}_{v \trianglelefteq e} = -(\widetilde{\mathcal{F}}^0_{u \trianglelefteq e} T^{(q)}_{vu})^* \widetilde{\mathcal{F}}_{v \trianglelefteq e} & \text{if } e = (v, u) \\ -\widetilde{\mathcal{F}}^*_{u \trianglelefteq e} \widetilde{\mathcal{F}}_{v \trianglelefteq e} = -\widetilde{\mathcal{F}}^T_{u \trianglelefteq e} \widetilde{\mathcal{F}}_{v \trianglelefteq e} & \text{if } e = \{u, v\} \\ 0 & \text{otherwise} \end{cases} \tag{2}$$

$$L^{\widetilde{\mathcal{F}}}_{uu} = \sum_{e \in \Gamma(u)} \widetilde{\mathcal{F}}^*_{u \trianglelefteq e} \widetilde{\mathcal{F}}_{u \trianglelefteq e}, \tag{3}$$

where $\Gamma(u)$ is the set of edges incident to $u$ regardless of their direction. Notice that, since with $q = \frac{1}{4}$ $(T^{(q)}_{vu})^* = -T^{(q)}_{uv}$, for a directed edge $e = (u, v)$ or $e = (v, e)$, $L^{\widetilde{\mathcal{F}}}_{uv}$ and $L^{\widetilde{\mathcal{F}}}_{vu}$ only differ by the sign of their imaginary part.

As one can see (the full derivation is reported in the appendix), when applied to a 0-cochain $x$, the Directed Sheaf Laplacian operator reads as follows for each $u \in V$:

$$L^{\widetilde{\mathcal{F}}}(x)_u = \underbrace{\sum_{e=(v,u)\in E} (\widetilde{\mathcal{F}}^0_{u \trianglelefteq e} T^{(q)}_{vu})^* (\widetilde{\mathcal{F}}_{u \trianglelefteq e} x_u - \widetilde{\mathcal{F}}_{v \trianglelefteq e} x_v)}_{\text{inflow}} \tag{4}$$

$$+ \underbrace{\sum_{e=(u,v)\in E} \widetilde{\mathcal{F}}^T_{u \trianglelefteq e} (\widetilde{\mathcal{F}}_{u \trianglelefteq e} x_u - \widetilde{\mathcal{F}}^0_{v \trianglelefteq e} T^{(q)}_{uv} x_v)}_{\text{outflow}} + \underbrace{\sum_{e=\{u,v\}\in E} \widetilde{\mathcal{F}}^T_{u \trianglelefteq e} (\widetilde{\mathcal{F}}_{u \trianglelefteq e} x_u - \widetilde{\mathcal{F}}_{v \trianglelefteq e} x_v)}_{\text{undirected}}.$$

We define the *normalized Directed Sheaf Laplacian* as:

$$L^{\widetilde{\mathcal{F}}}_N := \widetilde{D}^{-\frac{1}{2}} L^{\widetilde{\mathcal{F}}} \widetilde{D}^{-\frac{1}{2}}, \tag{5}$$

where $\widetilde{D} := \text{diag}(\widetilde{D}_1, \widetilde{D}_2, \ldots, \widetilde{D}_n)$ and, for all $u \in V$, $\widetilde{D}_u := \sum_{e \in \Gamma(u)} \widetilde{\mathcal{F}}^*_{u \trianglelefteq e} \widetilde{\mathcal{F}}_{u \trianglelefteq e}$.

## 3.2 Spectral properties of the Directed Sheaf Laplacian

The Directed Sheaf Laplacian enjoys several key spectral properties, which we now illustrate. The proofs of the theorems of this section and the next can be found in Appendix D. First, we show that both $L^{\widetilde{\mathcal{F}}}$ and $L^{\widetilde{\mathcal{F}}}_N$ are diagonalizable with a real spectrum and that their spectra are nonnegative:

**Theorem 1.** $L^{\widetilde{\mathcal{F}}}$ *is Hermitian and* $L^{\widetilde{\mathcal{F}}} \succeq 0$*, and the same holds for* $L^{\widetilde{\mathcal{F}}}_N$*.*

Next, we show that the spectrum of the Normalized Sheaf Laplacian is upper-bounded by 2:

**Theorem 2.** $L^{\widetilde{\mathcal{F}}_N} \preceq 2I$*.*

These theorems show that $L^{\widetilde{\mathcal{F}}}$ and $L^{\widetilde{\mathcal{F}}}_N$ enjoy the same spectral properties as the classical Laplacian matrix $L$ defined for undirected graphs. These are essential to define a principled convolutional operator by approximating the graph-Fourier transform of a graph signal with Chebyshev polynomials of the first kind of degree 1, as proposed by Kipf & Welling (2017) for the undirected case.

### 3.3 GENERALIZATION PROPERTIES OF THE DIRECTED SHEAF LAPLACIAN

First, we show that the Directed Sheaf Laplacian generalizes both the Sheaf Laplacian and the classical graph Laplacian:

**Theorem 3.** *If $G$ is undirected, $L^{\widetilde{\mathcal{F}}}$ coincides with the classical sheaf Laplacian $L^{\mathcal{F}}$ for any choice of $q \in \mathbb{R}$. Also, if the sheaf is trivial and $G$ is undirected and unweighted, $L^{\widetilde{\mathcal{F}}}$ coincides with the classical graph Laplacian $L$. If $G$ is directed and we set $q = 0$, $L^{\widetilde{\mathcal{F}}}$ coincides with the classical sheaf Laplacian associated with the undirected version of $G$.*

Let a *Trivial Directed Cellular Sheaf* be any Directed Cellular Sheaf with $d = 1$ where, for all directed edges $e = (u, v)$, $\widetilde{\mathcal{F}}_{u \trianglelefteq e} = 1$ and $\widetilde{\mathcal{F}}_{v \trianglelefteq e} = T_{uv}^{(q)}$. With the next theorem, we show that, for a given directed graph without weights, $L^{\widetilde{\mathcal{F}}}$ generalized the Magnetic Laplacian and, when choosing $q = \frac{1}{4}$, also the Sign-Magnetic Laplacian. The following holds:

**Theorem 4.** *Letting $G$ be a directed graph with unit weights, the Directed Sheaf Laplacian $L^{\widetilde{\mathcal{F}}}$ associated with a Trivial Directed Cellular Sheaf coincides with the Magnetic Laplacian $L^{(q)}$. In the special case where $q = \frac{1}{4}$, $L^{\widetilde{\mathcal{F}}}$ also coincides with the Sign-Magnetic Laplacian $L^{\sigma}$.*

It is well-known that the classical Laplacian matrix $L$ defined for an undirected graph can be equivalently defined as $L = D - A$ or $L = BB^T$, where $B \in \{-1, 0, 1\}^{n \times m}$ is the node-to-edge incidence matrix of the graph in which either of the two entries of each column has been arbitrarily multiplied by $-1$. While, to the best of our knowledge, no similar construction is known for the Magnetic Laplacian and the Sign-Magnetic Laplacian, with the following theorem we show that one such decomposition exists and that it can be obtained via the lens of our Directed Sheaf Laplacian (which generalizes both). Indeed, we have the following:

**Theorem 5.** *Let $G$ be a directed graph with unit weights. Assuming a Trivial Directed Cellular Sheaf, the conjugate transpose $\widetilde{\delta}^*$ of the Directed Coboundary Operator $\widetilde{\delta}$ boils down to the complex-valued node-to-edge incidence matrix $\widehat{B} \in \mathbb{C}^{n \times m}$ defined for an edge $e \in E$ incident to a vertex $u$:*

$$\widehat{B}_{ue} = \begin{cases} 1 & \text{if } e = (u, v) \text{ or } e = \{u, v\} \text{ with } u < v \\ -1 & \text{if } e = \{u, v\} \text{ with } u > v \\ -T_{uv}^{(q)} & \text{if } e = (v, u). \end{cases}$$

*It follows that $L^{(q)} = \widehat{B}\widehat{B}^*$. Also, when setting $q = \frac{1}{4}$, we have $L^{(\frac{1}{4})} = L^{\sigma} = \widehat{B}\widehat{B}^*$.*

Incidentally, this result leads to substantially simpler proofs of the positive semidefiniteness of both Laplacian matrices than those reported in their original papers.

## 4 THE DIRECTED SHEAF NEURAL NETWORK (DSNN)

The *sheaf diffusion* process on a graph $G$, introduced in Hansen & Gebhart (2020) as a generalization of the classical heat diffusion process that governs classical spectral-based GNNs Kipf & Welling (2017), follows the differential equation

$$\dot{X}(t) = -L_N^{\mathcal{F}} X(t),$$

where $X(t)$ is a time-dependent graph signal $X$. More precisely, $X_u$ is the stalk of each node $u \in V$, and it coincides with a matrix in $\mathbb{R}^{d \times f}$, where $d$ denotes the dimensionality of the vertex stalk and $f$ is the number of feature channels. $X$ is typically obtained starting from a matrix of node features of size $n \times f$ to which one applies a linear projection to obtain an $n \times (df)$ matrix, which is then reshaped to $(nd) \times f$.

By relying on our proposed Directed Sheaf Laplacian $L^{\widetilde{\mathcal{F}}}$, we introduce the *Directed Neural Sheaf Diffusion* process as the following generalization of the Neural Sheaf Diffusion process proposed by Bodnar et al. (2022):

$$\dot{X}(t) = -\sigma\left(L_N^{\widetilde{\mathcal{F}}}(t)\left(I_n \otimes W_1(t)\right)X(t)W_2(t)\right), \tag{6}$$

where $W_1 \in \mathbb{R}^{d \times d}$, $W_2 \in \mathbb{R}^{f \times f}$ are two time-dependent weight matrices and $\sigma$ is a nonlinear activation function.

We then define the **Directed Sheaf Neural Network** (DSNN) as the convolutional neural network whose convolution operator is obtained from the discretized version of Equation 6:

$$X^{(t+1)} = X^{(t)} - \sigma \left( L_N^{\widetilde{\mathcal{F}}(t)} \left( I_n \otimes W_1^{(t)} \right) X^{(t)} W_2^{(t)} \right), \tag{7}$$

where $X^{(t)}, X^{(t+1)} \in \mathbb{C}^{nd \times f}$.

The expressiveness of Eq. 7 is further enhanced by a learned parameter $\epsilon \in [-1, 1]^d$ by which we adjust the relative magnitude of the features in each component of a stalk. The update rule is thus:

$$X^{(t+1)} = \mathrm{diag}(1 + \varepsilon) X^{(t)} - \sigma \left( L_N^{\widetilde{\mathcal{F}}(t)} \left( I_n \otimes W_1^{(t)} \right) X^{(t)} W_2^{(t)} \right), \tag{8}$$

where $\varepsilon \in [-1, 1]^{nd}$ is obtained by concatenating $\epsilon$ $n$ times. As activation function $\sigma$, we adopt a complex extension of the *ReLU* function, defined for a given $z \in \mathbb{C}$, as

$$\sigma(z) = \begin{cases} z & \text{if } \Re(z) \geq 0, \\ 0 & \text{otherwise.} \end{cases}$$

This choice is consistent with previous work on complex-valued GNNs and HNNs, such as (Zhang et al., 2021b; Fiorini et al., 2024).

Finally, since our model operates in the complex domain, we project the output of the final layer to the real domain using an *unwind* operation. Given $X^{(\tau)} \in \mathbb{C}^{n \times c}$, the projection is defined as:

$$\mathrm{unwind}(X(\tau)) = (\Re(X(\tau)) \,\|\, \Im(X(\tau))) \in \mathbb{R}^{n \times 2c},$$

where $\tau$ is the last convolutional layer of the network, $\|$ denotes concatenation along the feature dimension, and $c$ is the output dimension.

**Learnable Sheaf Laplacian.** A key strength of SSNs is their ability to operate over richer structures, sheaves, rather than just the underlying graph. Since multiple sheaf structures can be associated with the same graph, effectively modeling the most suitable one is critical for a meaningful representation learning. In our proposed models, the restriction maps are learned end-to-end as a function of the input vertex features. Specifically, for each edge $e \in E$ with endpoints $u, v \in V$, each $d \times d$ matrix $\mathcal{F}_{u \trianglelefteq e}$ is parameterized as $\mathcal{F}_{u \trianglelefteq e} = \Phi(x_u \,\|\, x_v)$, where $x_v$ and $x_u$ denote the feature vectors of the nodes incident to $e$ and $\Phi$ is an MLP. The resulting vector is reshaped into a $d \times d$ matrix, thus obtaining the linear restriction map $\mathcal{F}_{u \trianglelefteq e}$.

**Connection with Neural Sheaf Diffusion.** The Neural Sheaf Diffusion process proposed by Bodnar et al. (2022) relies on the *Normalized Sheaf Laplacian* $L_N^{\mathcal{F}}$ instead of on our proposed *Directed Sheaf Laplacian* $L_N^{\widetilde{\mathcal{F}}}$ in Eq. 7. Since, as shown in Theorem 3, $L_N^{\mathcal{F}} = L_N^{\widetilde{\mathcal{F}}}$ when the graph in undirected, NSD is a special case of SNN when the graph is undirected.

**Computational Complexity.** Letting $f$ be the number of channels, assumed constant throughout the layers, we focus on a single convolutional layer. In the case of an undirected graph, where all restriction maps are real-valued, the complexity of DSNN is identical to the complexity of NSD, and reads $O\left(n(c^2 + d^3) + m(cd^2 + d^3)\right)$, which, with $d = 1$, coincides with the complexity of a classical spectral-based GNNs (Kipf & Welling, 2017), which is $\mathcal{O}(nc^2 + mc)$. In the experiments, we use $d \in \{2, 5\}$, which only introduces a small, constant overhead with no asymptotic impact. For a directed graph, the restriction maps are complex-valued, and thus, the stalks are complex-valued from layer 2 onward. This, though, only leads to an extra multiplicative cost of about 4, which is independent of the graph and size of the network and plays no role in the complexity of DSNN.

For the proof of the theorems in this section, and for additional details on DSNN's inference complexity, please refer to Appendix D and Appendix E, respectively. Computing times are reported in Table 4 in the appendix.

## 5 EXPERIMENTS

We compare DSNN against different state-of-the-art baselines on two complementary tasks, node classification and direction prediction, using both real-world and synthetic datasets. Following (Bodnar et al., 2022), we experiment with three types of $d \times d$ blocks in the Directed Sheaf Laplacian $L^{\widetilde{\mathcal{F}}}$,

*diagonal*, *orthogonal*, and *general*, which lead to three variants of DSNN: Diag-DSNN, O(d)-DSNN, Gen-DSNN. From Definition 1, the normalized directed sheaf Laplacian $L_N^{\widetilde{\mathcal{F}}(t)}$ depends on a parameter $q$. As $L_N^{\widetilde{\mathcal{F}}(t)}$ is used in the convolutional layer (see Equation 7), we treat $q$ as a hyperparameter in this work. In the tables, the best results are reported in **boldface** and the second-best are underlined. The datasets and code we used are available on GitHub (see Appendix B). Further details on our experiments are reported in Appendix F, G.

**Baselines.** We compare DSNN against a large set of GNN and SNN baselines from five categories: *i)* classical GNN models: GCN (Kipf & Welling, 2017), GAT (Veličković et al., 2018); *ii)* GNN models designed for heterophilic graphs: Geom-GCN Pei et al. (2020), H2GCN (Zhu et al., 2020), GPRGNN (Chien et al., 2021), FAGCN (Bo et al., 2021), GGCN (Yan et al., 2022); HSGNN Lu et al. (2024) *iii)* GNN models that address the oversmoothing problem: GCNII (Chen et al., 2020); *iv)* GNN models that incorporate edge directionality: DiGCL (Tong et al., 2021), MagNet (Zhang et al., 2021b), SigMaNet (Fiorini et al., 2023), DirGNN (Rossi et al., 2024), HaarNet (Badea & Dumitrescu, 2025), CAGN Xu et al. (2025); *v)* SNN models: NSD (Bodnar et al., 2022).

**Real-world datasets.** The `Texas`, `Wisconsin`, `Cornell`, and `Film` datasets are provided by Pei et al. (2020); `Citeseer`, `PubMed`, and `Cora` by Yan et al. (2022); `Squirrel`, `Chameleon`, `Roman-Empire`, and `Questions` by Platonov et al. (2023); and `Telegram` by Bovet & Grindrod (2020). Since GNNs are known to struggle on heterophilic graphs (where neighbors often have different labels), we evaluate DSNN across datasets with a wide range of edge homophily coefficients.

**Node classification on real-world datasets.** We follow the evaluation protocol of Bodnar et al. (2022) using 10 splits. For Texas, Wisconsin, Film, Cornell, Citeseer, Pubmed, and Cora, we rely on the fixed splits provided by Yan et al. (2022) and report results on all 10 predefined splits. For Chameleon, Squirrel, Roman-Empire, and Questions, we adopt the splitting strategy from Platonov et al. (2023). For Telegram, we use the split introduced in Zhang et al. (2021b). We report mean±std accuracy for all datasets, and ROC AUC for `Questions`, which, as noted by Platonov et al. (2023), is highly imbalanced with 97% of the users belonging to the majority class. Table 1 shows that DSNN attains the best results on 10/12 benchmarks, spanning both heterophilic and homophilic graphs. In particular, `Texas`, `Wisconsin`, `Film`, `Chameleon`, `Cornell`, `Citeseer`, `PubMed`, `Telegram`, `Roman-Empire`, and `Questions` are won by a DSNN variant, while the strongest baselines narrowly lead only on `Squirrel` and `Cora` and only by a small margin (0.22 and 0.79, respectively). Compared to the SNN baseline (NSD), adding directionality within the sheaf systematically helps, as DSNN surpasses NSD on the majority of datasets, with notable margins on `Questions`, `Texas`, `Telegram`, and `Roman-Empire`. Relative to direction-aware GNNs (DirGNN, SigMaNet, Mag-Net), DSNN delivers consistently stronger performance on 10/12 datasets, suggesting that combining cellular-sheaf expressivity with an explicit oriented Laplacian yields a more effective message-passing bias. Overall, these results indicate that learning direction-aware restriction maps within a sheaf framework provides robust gains across many graph regimes, especially in challenging heterophilic settings. See Appendix H for the $q$ values of DSNN compared to the ones of MagNet, Appendix I for an analysis of $d$ values in DSNN, and Appendix L for a preliminary analysis of $q$ when set as a learnable parameter.

**Node classification on synthetic datasets.** We further investigate the role of directionality by comparing DSNN against NSD on a set of synthetic graphs generated using the Direct Stochastic Block Model (DSBM) (He et al., 2022). For this experiment, the DBSM datasets are generated with $n = 2500$ nodes, $C = 5$ classes, intra-cluster density $\alpha_{ii} = 0.1$, probability of an edge taking a certain direction $\beta_{ij} = 0.2$, and with an increasing inter-cluster density $\alpha_{ij} \in \{0.05, 0.08, 0.1\}$. Features are intentionally minimal (scalar in/out–degree sums). We run the experiments 10 times per dataset with a 80%/5%/15% training/validation/testing split, and report the mean accuracy and standard deviation. Across all settings in Table 2, the three DSNN variants achieve near-perfect accuracy (typically 96–99%; best: 99.14% at $\alpha_{ij} = 0.10$), substantially outperforming NSD ($\sim 20\%$, i.e., $\frac{1}{5}$ chance for $C = 5$). Direction-aware GNNs trend upward with density, but still trail DSNN: MagNet rises from 78.64% to 91.58%, and SigMaNet narrows the gap at higher density (up to 98.60%) yet remains below DSNN on average. DirGNN shows instability with a large standard error, suggesting sensitivity to directed community structure under our minimal features. Together, these results indicate that explicitly encoding orientation in the sheaf Laplacian yields a decisive advantage when communities are directionally biased, even when node features carry little information. Consistent with Table 11 (in the appendix), setting $q = 0$—i.e., discarding edge orientation inside the sheaf—leads to a drop in

Table 1: Node classification results. ROC AUC curve is reported on `Questions` while mean and standard deviation accuracy is reported for all other datasets

| Model | Roman-Empire | Texas | Wisconsin | Film | Squirrel | Chameleon | Cornell | Telegram | Citeseer | Pubmed | Cora | Questions |
|---|---|---|---|---|---|---|---|---|---|---|---|---|
| Homoph. lvl | 0.05 | 0.11 | 0.21 | 0.22 | 0.22 | 0.23 | 0.30 | 0.32 | 0.74 | 0.80 | 0.81 | 0.84 |
| # Nodes | 22,662 | 183 | 251 | 7,600 | 2,223 | 890 | 183 | 245 | 3,327 | 18,717 | 2,708 | 48,921 |
| # Edges | 32,927 | 295 | 466 | 26,752 | 46,998 | 8,854 | 280 | 8,912 | 4,676 | 44,327 | 5,278 | 153,540 |
| # Classes | 18 | 5 | 5 | 5 | 5 | 5 | 5 | 4 | 6 | 3 | 7 | 2 |
| **Diag-DSNN** | 90.40±0.31 | **88.65±4.95** | **90.20±4.02** | 38.34±1.01 | 45.37±2.21 | 46.84±4.03 | **87.84±5.70** | 94.42±3.03 | 79.80±1.49 | **90.23±0.44** | 87.36±1.41 | 79.08±0.72 |
| **O(d)-DSNN** | **92.08±0.24** | 87.57±4.04 | 89.80±3.82 | 37.37±0.98 | 44.54±2.26 | 45.36±3.29 | 87.30±7.26 | 94.62±2.24 | 77.28±1.63 | 90.05±0.55 | 87.30±1.62 | 79.24±0.68 |
| **Gen-DSNN** | **92.08±0.36** | 87.57±5.43 | 89.22±3.31 | **38.40±0.75** | 45.34±1.69 | **47.16±3.54** | 87.84±6.86 | **94.81±2.28** | **79.88±1.21** | 90.17±0.44 | 87.58±0.72 | **79.55±0.67** |
| Diag-NSD | 83.20±0.47 | 85.67±6.95 | 88.63±2.75 | 37.79±1.01 | 45.52±2.22 | 46.55±3.03 | 86.49±7.35 | 92.11±3.38 | 89.42±0.43 | 87.14±1.06 | 75.82±1.05 |
| O(d)-NSD | 83.67±0.34 | 85.95±5.51 | 89.41±4.74 | 37.81±1.15 | 45.59±2.23 | 46.26±3.11 | 84.86±4.71 | 91.53±2.46 | 76.70±1.57 | 89.49±0.40 | 86.90±1.13 | 77.19±1.37 |
| Gen-NSD | 83.80±0.50 | 82.97±5.13 | 89.21±3.84 | 37.80±1.22 | 45.31±2.05 | 45.60±3.36 | 85.68±6.51 | 91.73±2.44 | 76.32±1.65 | 89.33±0.35 | 87.30±1.15 | 77.36±1.32 |
| HaarNet | 85.42±0.43 | 77.57±4.18 | 71.56±6.69 | 36.38±1.01 | 40.52±3.14 | 42.43±3.98 | 73.91±7.57 | 91.12±3.69 | 76.51±1.64 | 88.39±0.61 | 82.68±1.54 | 75.01±0.94 |
| CAGN | OOM | 75.67±7.15 | 84.11±4.51 | 34.86±1.06 | 35.38±0.99 | 39.79±3.89 | 73.79±7.25 | 86.73±3.69 | 73.64±2.81 | OOM | 86.23±1.05 | OOM |
| DiGCL | 52.71±0.32 | 57.56±5.15 | 65.50±4.23 | 29.38±0.73 | 38.90±1.78 | 41.71±2.20 | 62.16±5.12 | 80.57±2.25 | 77.42±0.14 | 80.97±0.7 | 76.12±1.04 | OOM |
| DirGnn | 91.23±0.32 | 74.22±3.97 | 71.37±6.57 | 29.30±1.22 | 44.48±1.94 | 45.56±3.36 | 61.46±3.63 | 92.81±2.07 | 76.09±1.53 | 85.14±0.44 | 86.20±1.18 | 76.57±0.86 |
| SigMaNet | 85.60±0.29 | 78.92±4.49 | 80.21±5.07 | 36.59±0.55 | 40.89±1.97 | 40.98±3.88 | 73.53±5.91 | 86.12±3.49 | 74.35±0.96 | 88.35±0.64 | 85.51±1.14 | 76.95±0.95 |
| MagNet | 88.07±0.27 | 79.46±8.13 | 81.18±2.80 | 36.51±0.96 | 41.04±1.84 | 43.82±4.56 | 75.99±5.59 | 87.62±2.92 | 77.21±1.69 | 88.47±0.54 | 86.32±1.39 | 75.66±0.63 |
| H2SGNN | 69.59±0.45 | 72.70±8.83 | 78.23±5.22 | 36.75±1.33 | 37.09±1.21 | 41.14±3.60 | 74.05±5.94 | 62.69±3.95 | 77.17±1.36 | 86.94±0.42 | 82.41±1.42 | 74.20±0.65 |
| GGCN | 76.25±0.48 | 84.86±4.55 | 86.86±3.29 | 37.54±1.56 | 40.75±2.44 | 39.71±3.25 | 85.59±4.12 | 73.55±5.16 | 77.14±1.45 | 89.15±0.37 | 87.95±1.05 | 74.19±1.01 |
| H2GCN | 60.11±0.52 | 84.86±7.23 | 87.65±4.98 | 35.70±1.00 | 37.77±1.92 | 42.07±4.13 | 82.70±5.28 | 88.27±3.89 | 77.11±1.57 | 89.49±0.38 | 87.87±1.20 | 75.30±1.35 |
| GPRGNN | 68.85±0.27 | 78.38±4.36 | 82.94±4.21 | 34.63±1.22 | 36.62±2.28 | 40.67±2.89 | 80.27±5.81 | 74.23±6.45 | 77.13±1.67 | 87.54±0.38 | 87.95±1.18 | 75.42±1.29 |
| FAGCN | 74.75±0.72 | 82.43±6.89 | 82.94±7.95 | 34.87±1.25 | 41.08±2.27 | 41.90±2.72 | 79.19±9.79 | 80.77±7.79 | 77.10±1.81 | 90.21±0.36 | 88.17±1.24 | 76.40±2.01 |
| GCNII | 83.70±0.51 | 77.57±3.83 | 80.39±3.40 | 37.44±1.30 | 42.22±2.13 | 43.76±2.49 | 77.86±3.79 | 89.03±3.95 | 77.33±1.48 | 90.15±0.43 | 88.37±1.25 | 78.03±0.84 |
| GCN | 73.69±0.74 | 55.14±5.16 | 51.76±3.06 | 27.32±1.10 | 39.47±1.47 | 40.89±4.12 | 60.54±5.30 | 73.43±5.81 | 76.50±1.36 | 88.42±0.50 | 86.98±1.27 | 74.61±0.82 |
| GAT | 69.07±0.83 | 52.16±6.63 | 49.41±4.09 | 27.44±0.89 | 35.62±2.06 | 39.21±3.08 | 61.89±5.05 | 72.61±7.50 | 76.55±1.23 | 87.30±1.10 | 86.33±0.48 | 76.56±0.93 |
| MLP | 64.94±0.62 | 80.81±4.75 | 85.29±3.31 | 36.53±0.70 | 40.45±1.41 | 42.79±3.80 | 81.89±6.40 | 46.34±5.47 | 74.02±1.90 | 87.16±0.37 | 75.69±2.00 | 71.23±0.94 |

Table 2: Node classification results on synthetic dataset. Performance comparison with different $\alpha_{ij}$ values can be seen on the tables.

| Model / $\alpha$ | 0.05 | 0.08 | 0.10 |
|---|---|---|---|
| **Diag-DSNN** | **98.34±0.72** | 97.22±0.58 | **99.14±0.36** |
| **O(d)-DSNN** | 97.28±0.68 | **98.42±0.61** | 98.80±0.27 |
| **Gen-DSNN** | 96.64±0.86 | 98.10±0.65 | 98.68±0.45 |
| Diag-NSD | 20.64±1.84 | 21.42±1.05 | 20.58±1.20 |
| O(d)-NSD | 20.15±1.45 | 20.57±1.25 | 20.41±0.85 |
| Gen-NSD | 20.20±1.08 | 20.42±1.49 | 20.51±0.89 |
| HaarNet | 98.18±0.53 | 98.25±0.33 | 98.82±0.54 |
| CAGN | 92.71±7.65 | 93.31±7.75 | 96.96±5.86 |
| DiGCL | 30.13±5.12 | 31.54±6.87 | 24.73±9.12 |
| MagNet | 78.64±1.29 | 87.52±1.30 | 91.58±1.04 |
| SigMaNet | 87.44±0.99 | 96.14±0.64 | 98.60±0.31 |
| DirGNN | 83.96±7.91 | 83.32±12.92 | 83.16±17.31 |

| Model / $\alpha$ | 0.05 | 0.08 | 0.10 |
|---|---|---|---|
| H2SGNN | 20.32±0.88 | 21.08±2.16 | 20.28±0.53 |
| GGCN | 20.14±0.31 | 20.02±0.06 | 20.01±2.78 |
| H2GCN | 20.34±1.49 | 20.68±1.23 | 20.28±0.54 |
| GPRGNN | 40.02±0.06 | 40.14±0.29 | 38.96±0.73 |
| FAGCN | 20.46±0.92 | 20.48±1.02 | 20.24±0.51 |
| GCNII | 20.84±1.31 | 20.71±0.83 | 20.73±1.41 |
| GAT | 23.34±3.13 | 21.98±2.09 | 21.58±1.84 |
| GCN | 20.06±0.18 | 20.46±0.22 | 20.01±1.01 |
| MLP | 20.28±0.69 | 20.31±0.93 | 20.28±0.45 |

performance, whereas $q > 0$ restores the gains. This confirms that DSNN's improvements stem from using directionality inside the restriction maps rather than from the increased capacity of the network.

**Direction prediction on the real-world datasets.** We further test the performance of DSNN on the *direction prediction* task: a binary classification task where the model is asked to predict whether $(u, v) \in E$ or $(v, u) \in E$. Following Zhang et al. (2021b), we split edges into 15% test, 5% validation, and the rest for training, and perform 10-fold cross-validation while preserving graph connectivity. As Table 3 shows, DSNN attains state-of-the-art performance on 6/10 datasets (`Texas`, `Wisconsin`, `Cornell`, `Citeseer`, `Squirrel`, `Pubmed`, `Questions`). On the remaining ones, DSNN is a close runner-up: on `Cora` it is within 0.01 accuracy of the best method, and on `Film` it trails the strongest direction-aware GNN by 0.18. We observe that using a nonzero $q$ is beneficial on most benchmarks. In the appendix (Table 10), the best settings adopt $q > 0$ in most model–dataset combinations, corroborating that explicitly encoding edge orientation inside the directed sheaf Laplacian $L^{\widetilde{\mathcal{F}}}$ improves disambiguation of $(u, v)$ versus $(v, u)$. Overall, these results mirror our node-classification findings: injecting direction into the sheaf (via complex restriction maps and $L^{\widetilde{\mathcal{F}}}$) yields a robust advantage on tasks where edge orientation is key.

**Training and inference time.** Wall-clock time (training+testing, averaged over 10 folds, 1k epochs) and peak GPU memory are reported in Tables 4 and 5 (see Appendix E for more details). DSNN comes out as consistently faster than generic direction-aware GNNs, but (as expected) incurs a moderate constant-factor overhead vs. NSD due to complex-valued arithmetic. On small-to-medium graphs, DSNN runs in ≈8–10 s per experiment (vs. 6.5–7.8 s for NSD), while on denser or larger graphs times are 19–33 s for DSNN (vs. 10–16 s for NSD) and 107 s for DSNN vs 47.5 s for NSD on the largest benchmark (`Questions`). Memory usage shows a similar pattern: DSNN peaks at

Table 3: Results on direction prediction. Mean and standard deviation accuracy are reported.

| Model | Texas | Wisconsin | Cornell | Cora | Citeseer | Film | Squirrel | Chameleon | Pubmed | Questions |
|---|---|---|---|---|---|---|---|---|---|---|
| **Diag-DSNN** | 93.36±2.64 | 87.18±4.02 | 89.56±4.41 | 82.54±1.02 | 85.10±0.79 | 81.20±0.62 | 95.17±0.24 | 92.04±0.73 | 95.41±0.20 | **90.48±0.11** |
| **O(d)-DSNN** | **94.55±4.61** | **88.31±3.38** | **91.06±3.96** | **82.98±1.08** | **85.44±0.82** | 81.18±0.52 | **95.53±0.31** | 92.52±0.43 | 95.23±0.23 | 90.16±0.13 |
| **Gen-DSNN** | 93.62±3.56 | 87.84±4.04 | 90.83±3.73 | 82.71±1.35 | 85.01±1.01 | 81.20±0.61 | 95.27±0.26 | 92.41±0.49 | **95.56±0.19** | 90.23±0.15 |
| Diag-NSD | 88.42±4.64 | 85.66±3.76 | 85.78±3.39 | 82.76±1.19 | 85.05±1.43 | 80.96±0.52 | 91.56±0.66 | 90.12±1.44 | 95.25±0.18 | 90.29±0.15 |
| O(d)-NSD | 88.53±4.44 | 85.64±4.13 | 85.38±4.97 | 82.71±1.14 | 85.19±0.97 | 81.37±0.68 | 92.19±1.23 | 88.22±0.74 | 95.36±0.22 | 90.12±0.11 |
| Gen-NSD | 89.59±4.14 | 85.89±4.33 | 85.58±4.86 | 82.58±0.73 | 85.06±0.85 | 80.09±0.59 | 88.96±0.45 | 90.98±0.92 | 94.99±0.23 | 90.21±0.12 |
| HaarNet | 86.15±5.44 | 86.79±2.31 | 85.55±6.23 | 82.53±0.79 | 85.13±0.89 | 80.86±0.68 | 95.04±0.37 | 92.11±0.58 | 95.13±0.27 | 90.10±0.18 |
| CAGN | 87.34±3.89 | 85.79±3.02 | 84.81±4.31 | 81.94±0.95 | 84.73±0.67 | 80.97±0.77 | 95.08±0.19 | **94.84±0.50** | 95.18±0.32 | OOM |
| Dir-GNN | 88.35±4.66 | 86.13±3.91 | 85.59±4.95 | **82.99±0.82** | 84.31±1.51 | 80.56±0.42 | 96.23±0.24 | 93.54±0.63 | 95.22±0.21 | 90.03±0.17 |
| SigMaNet | 89.37±3.65 | 86.53±3.79 | 85.37±4.51 | 81.98±0.78 | 84.29±0.98 | 80.84±0.65 | 94.98±0.44 | 92.01±0.59 | 95.05±0.22 | 90.05±0.14 |
| MagNet | 88.94±3.96 | 86.65±3.34 | 85.77±3.52 | 82.25±0.84 | 84.64±1.01 | 81.01±0.51 | 94.99±0.41 | 92.09±0.58 | 95.28±0.31 | 90.03±0.21 |
| H2SGNN | 90.19±4.07 | 86.62±3.24 | 86.36±3.41 | 81.21±1.06 | 84.69±0.88 | 80.32±0.75 | 94.16±0.42 | 91.14±0.53 | 94.80±0.22 | 88.98±0.25 |
| GGCN | 89.59±4.08 | 85.67±2.82 | 86.16±3.26 | 82.73±0.73 | 84.93±0.63 | 81.29±0.68 | 93.93±0.55 | 91.55±0.72 | 94.98±0.20 | 90.17±0.18 |
| H2GCN | 87.64±6.17 | 84.46±4.13 | 84.59±7.18 | 81.42±0.91 | 84.22±0.98 | 80.82±0.56 | 92.18±0.26 | 89.96±0.68 | 94.69±0.26 | 88.98±0.25 |
| GPRGNN | 87.58±4.67 | 84.94±3.67 | 84.41±5.29 | 81.87±1.03 | 84.56±1.45 | 80.49±0.65 | 93.12±0.46 | 89.55±0.66 | 94.82±0.55 | 89.85±0.41 |
| FAGCN | 88.15±5.74 | 86.26±4.78 | 83.06±7.29 | 82.15±0.94 | 84.68±1.42 | 80.61±0.41 | 94.08±0.44 | 91.28±1.17 | 95.05±0.47 | 89.15±0.31 |
| GCNII | 89.39±4.81 | 85.66±2.87 | 84.43±4.57 | 82.59±0.95 | 84.98±0.87 | 81.02±0.74 | 95.38±0.40 | 92.06±0.67 | 95.09±0.21 | 90.17±0.18 |
| GAT | 89.15±5.52 | 85.77±3.62 | 85.01±5.74 | 82.14± 1.27 | 84.54±1.11 | 78.59±1.01 | 95.07±0.26 | 92.25±0.90 | 95.04±0.21 | 87.73±0.99 |
| GCN | 86.90±3.68 | 86.90±3.68 | 76.20±6.33 | 81.78±1.04 | 83.63±1.05 | 77.61±0.47 | 94.10±0.25 | 90.70±0.56 | 86.95±0.56 | 77.86±0.35 |
| MLP | 90.21±4.35 | 86.61±4.30 | 87.12±4.06 | 82.54±1.09 | 84.98±0.82 | **81.38±0.66** | 93.42±0.32 | 90.08±0.69 | 95.03±0.18 | 90.01±0.12 |

404–4462 MiB versus 384–3360 MiB for NSD (i.e., typically a 5–35% increase). Importantly, both methods share the same asymptotic complexity, as shown in Appendix E.

Table 4: Training and testing time (s), averaged over 10 folds and 1,000 epochs.

| Model | Texas | Wisconsin | Film | Squirrel | Chameleon | Cornell | Citeseer | Pubmed | Cora | Roman-Empire | Questions | Telegram |
|---|---|---|---|---|---|---|---|---|---|---|---|---|
| **DSNN** ($q=0.25$) | 8.23±0.11 | 8.30±0.13 | 19.20±0.08 | 25.79±0.12 | 9.75±0.18 | 8.27±0.15 | 9.76±0.10 | 33.15±0.05 | 9.10±0.21 | 28.43±0.076 | 106.99±0.09 | 9.58 ± 0.32 |
| **DSNN** ($q=0.15$) | 8.23±0.24 | 8.24±0.10 | 19.25±0.08 | 25.66±0.22 | 9.77±0.12 | 8.33±0.19 | 9.81±0.17 | 33.14±0.05 | 9.10±0.18 | 28.87±0.045 | 106.50±0.06 | 9.61± 0.24 |
| **DSNN** ($q=0$) | 8.20±0.15 | 8.42±0.17 | 19.20±0.09 | 25.79±0.11 | 9.91±0.26 | 8.30±0.17 | 9.72±0.10 | 33.22±0.07 | 9.01±0.19 | 28.22±0.086 | 107.10±0.13 | 9.46 ± 0.22 |
| NSD | 6.59±0.15 | 6.64±0.10 | 10.73±0.14 | 14.43±0.12 | 7.79±0.24 | 6.59±0.10 | 7.50±0.17 | 16.32±0.08 | 7.39±0.13 | 14.15±0.084 | 47.51±0.12 | 7.69 ± 0.33 |

Table 5: Peak GPU memory (MiB).

| Model | Texas | Wisconsin | Film | Squirrel | Chameleon | Cornell | Citeseer | Pubmed | Cora | Roman-Empire | Questions | Telegram |
|---|---|---|---|---|---|---|---|---|---|---|---|---|
| **DSNN** | 404 | 408 | 1078 | 1362 | 598 | 404 | 830 | 1744 | 606 | 1100 | 4462 | 414 |
| NSD | 384 | 400 | 920 | 1180 | 554 | 384 | 588 | 1338 | 522 | 730 | 3360 | 390 |

## 6 Conclusions

We introduced the Directed Cellular Sheaf, from which we derived the Directed Sheaf Laplacian $L^{\widetilde{\mathcal{F}}}$. By encoding the edge direction in its imaginary components, $L^{\widetilde{\mathcal{F}}}$ carries a directional inductive bias thanks to which we obtain a convolution operator implementing a message-passing scheme capable of handling asymmetric interactions. We embedded such an operator in the Directed Sheaf Neural Network (DSNN). Our theoretical results showed that DSNN generalizes several well-established graph-learning models, including NSD, MagNet, and SigMaNet. Empirically, DSNN exhibits strong performance across both real-world and synthetic datasets, consistently outperforming both traditional GNNs and SNNs. This demonstrates that DSNN's explicit treatment of directionality leads to superior generalization, particularly in heterophilic graph settings. Future works include extending our approach to temporal graphs to broaden its applicability to dynamic settings.

## Acknowledgement of Support

Stefano Fiorini's work was carried out during a research visit at the University of Cambridge, supported by the ELSA – European Lighthouse on Secure and Safe AI project, and funded by the European Union under grant agreement No. 101070617. Stefano Coniglio's work was partially supported by the European Union under Next Generation EU — the Italian National Recovery and Resilience Plan (PNRR), PRIN 2022 PNRR (project code P20227CTY3, CUP D53D23018800001), project title "HEXAGON: Highly-specialized EXact Algorithms for Grid Operations at the National level".

## ETHICS STATEMENT

We adhere to the ICLR Code of Ethics. This work studies sheaf-based learning on graphs and does not involve human subjects, personally identifiable information, or sensitive attributes. The real-world benchmarks we use are standard public datasets (with licenses referenced in Appendix B); our synthetic graphs are generated procedurally as described in Section 5 and Appendix F. We release a code repository under a permissive license to facilitate verification and reuse.

As with any graph-learning technique, downstream applications to human-centered data could raise concerns around privacy, fairness, or surveillance. Our contribution is methodological and evaluated on public or synthetic data; nevertheless, we encourage practitioners to assess domain-specific risks, follow applicable regulations, and adopt appropriate safeguards (e.g., data minimization, bias checks) when deploying such models.

## REPRODUCIBILITY STATEMENT

We took several steps to support reproducibility. All model components, including the Directed Sheaf Laplacian $L^{\widetilde{\mathcal{F}}}$, training objectives, and update rules, are fully specified in the main text, with additional implementation details in Appendices E and G. Dataset sources, preprocessing, and synthetic graph generation (parameter grids for $\alpha_{ij}$, $\beta_{ij}$, and $q$) are documented in Appendix F. We report splits, evaluation protocols, and hyperparameter search spaces in Section 5 and Appendix G, and we include hardware information and training schedules.

Code and scripts to reproduce all tables (including random seeds and configuration files) accompanies this submission (Appendix B). Note that exact bitwise determinism can depend on backend/library settings (e.g., CUDA), but we fix seeds and document any sources of nondeterminism.

## LLM USAGE STATEMENT

We did *not* use large language models (LLMs) for deriving, checking, or producing any proofs or theoretical results in this paper. All theorems and proofs were conceived, implemented, and validated by the authors.

LLMs were used only as general-purpose assistants for: (i) light prose edits (clarity/grammar) and (ii) minor LaTeX refactoring (e.g., formatting environments). All such edits were manually reviewed. No human-subject data, personally identifiable information, or proprietary datasets were provided to any LLM, and all experimental code runs independently of LLM services.

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

## A  NOTATION SUMMARY

Table 6 lists the notation used in the main paper.

Table 6: Notation

| Symbol | Meaning |
|---|---|
| $G = (V, E)$ | Directed graph with vertex set $V$, edge set $E$ |
| $n = |V|, \ m = |E|$ | Number of vertices and edges |
| $A \in \{0, 1\}^{n \times n}$ | Adjacency matrix of $G$ |
| $A_s = \frac{1}{2}(A + A^\top)$ | Symmetrized adjacency matrix |
| $\mathbf{i}$ | Imaginary unit ($\mathbf{i}^2 = -1$) |
| $T^{(q)} = \exp(\mathbf{i}\, 2\pi q(A - A^\top))$ | Phase–direction matrix (Magnetic term) |
| $q \in \mathbb{R}$ | Directionality parameter |
| $\mathcal{F}(u) \in \mathbb{C}^d$ | Stalk (vector space) at vertex $u$ |
| $\mathcal{F}(e) \in \mathbb{C}^d$ | Stalk at edge $e$ |
| $\mathcal{F}_{u \trianglelefteq e}$ | Restriction map from node $u$ to edge $e$ |
| $\mathcal{F}^0_{u \trianglelefteq e}$ | Real base map (before applying $T^{(q)}$) |
| $C^0(G; \mathcal{F}) = \bigoplus_{u \in V} \mathcal{F}(u)$ | Space of 0-cochains |
| $C^1(G; \mathcal{F}) = \bigoplus_{e \in E} \mathcal{F}(e)$ | Space of 1-cochains |
| $\widetilde{\delta} : C^0 \to C^1$ | Directed coboundary operator |
| $\widetilde{\delta}(x)_e = \mathcal{F}_{u \trianglelefteq e} x_u - \mathcal{F}_{v \trianglelefteq e} x_v$ | Action on cochain $x$ |
| $L^{\widetilde{\mathcal{F}}} = \widetilde{\delta}^* \widetilde{\delta}$ | Directed Sheaf Laplacian (DSL) |
| $L^{\widetilde{\mathcal{F}}}_N = \widetilde{D}^{-1/2} L^{\widetilde{\mathcal{F}}} \widetilde{D}^{-1/2}$ | Normalized DSL |
| $\widetilde{D}_u = \sum_{e \in \Gamma(u)} \mathcal{F}^*_{u \trianglelefteq e} \mathcal{F}_{u \trianglelefteq e}$ | Degree block for node $u$ |
| $\widetilde{D} = \mathrm{diag}(\widetilde{D}_1, \dots, \widetilde{D}_n)$ | Block degree matrix |
| $\Gamma(u)$ | Set of edges incident to $u$ (ignoring direction) |
| $E^0, E^1$ | Sets of undirected and directed edges |
| $X^{(t)} \in \mathbb{C}^{nd \times f}$ | Node feature cochain at layer $t$ |
| $W_1^{(t)}, W_2^{(t)}$ | Learnable weight matrices at layer $t$ |
| $\varepsilon \in [-1, 1]^d$ | Learned scaling of stalk components |
| $\sigma(\cdot)$ | Complex ReLU activation |
| $\mathrm{unwind}(Z)$ | Concatenation of real and imaginary parts of $Z$ |
| $\widehat{B}$ | Complex incidence matrix (for trivial sheaf) |
| $L^{(q)}$ | Magnetic Laplacian |
| $L^\sigma$ | Sign-Magnetic Laplacian |

## B  CODE REPOSITORY AND LICENSING

The code written for this research work is available at `https://github.com/hakanaktas0/DSNN` and freely distributed under the Apache 2.0 license.[1]

The `Cora`, `Citeseer`, and `PubMed` datasets are available at `https://linqs.org/datasets/` `Citeseer`, `PubMed`, and `Cora` are sourced from (Yan et al., 2022). The `Squirrel`, `Chameleon`, `Roman-Empire` and `Questions` datasets come from Platonov et al. (2023). `Telegram` is provided by Bovet & Grindrod (2020).

---

[1] `https://www.apache.org/licenses/LICENSE-2.0`

Regarding the models used in this paper, we rely on publicly available implementations from the following sources:

- **MLP, GCN, GAT, GGCN, GCNII, GPRGNN:** `https://github.com/Yujun-Yan/Heterophily_and_oversmoothing` with MIT license.
- **H2GCN:** `https://github.com/Godofnothing/HeterophilySpecificModels/tree/main/H2GCN`.
- **FAGCN:** `https://github.com/Godofnothing/HeterophilySpecificModels/tree/main/FAGCN`.
- **MagNet:** `https://github.com/matthew-hirn/magnet` with Apache License 2.0.
- **SigMaNet:** `https://github.com/Stefa1994/SigMaNet` with Apache License 2.0.
- **DirGNN:** `https://github.com/emalgorithm/directed-graph-neural-network` with Apache License 2.0.
- **NSD:** `https://github.com/twitter-research/neural-sheaf-diffusion` with Apache License 2.0.
- **HaarNet:** `https://github.com/theodorbadea/Haar-Laplacian/tree/main`.
- **CAGN:** `https://github.com/LaZuiYa/CAGN/tree/main` with MIT license.
- **H2SGNN:** `https://github.com/Lukangkang123/H2SGNN/tree/main`
- **DiGCL:** `https://github.com/flyingtango/DiGCL` with MIT license.

## C  DERIVATION OF THE EQUATION OF $L^{\widetilde{\mathcal{F}}}$

Since, by construction, $L^{\widetilde{\mathcal{F}}} = \widetilde{\delta}^* \widetilde{\delta}$, the following equation holds:

$$L^{\widetilde{\mathcal{F}}}(x)_u = \underbrace{\sum_{e \in \Gamma(u)} \widetilde{\mathcal{F}}^*_{u \trianglelefteq e} \widetilde{\mathcal{F}}_{u \trianglelefteq e} x_u}_{\text{self-loop}} - \underbrace{\sum_{e=(v,u) \in E} (\mathcal{F}^0_{u \trianglelefteq e} T^{(q)}_{vu})^* \mathcal{F}_{v \trianglelefteq e} x_v}_{\text{inflow}}$$

$$- \underbrace{\sum_{e=(u,v) \in E} \mathcal{F}^T_{u \trianglelefteq e} \mathcal{F}^0_{v \trianglelefteq e} T^{(q)}_{uv} x_v}_{\text{outflow}} - \underbrace{\sum_{e=\{u,v\} \in E} \mathcal{F}^T_{u \trianglelefteq e} \mathcal{F}_{v \trianglelefteq e} x_v}_{\text{undirected}}.$$

For a directed graph $G$, the $uu$ component of $L^{\widetilde{\mathcal{F}}}$ can be rewritten as follows:

$$L^{\widetilde{\mathcal{F}}}_{uu} = \sum_{e \in \Gamma(u)} \widetilde{\mathcal{F}}^*_{u \trianglelefteq e} \widetilde{\mathcal{F}}_{u \trianglelefteq e} = \sum_{e=(u,v) \in E} \widetilde{\mathcal{F}}^T_{u \trianglelefteq e} \widetilde{\mathcal{F}}_{u \trianglelefteq e}$$

$$+ \sum_{e=(v,u) \in E} (\widetilde{\mathcal{F}}^0_{u \trianglelefteq e})^T (T^{(q)}_{vu})^* \widetilde{\mathcal{F}}^0_{u \trianglelefteq e} T^{(q)}_{vu} + \sum_{e=\{u,v\} \in E} \widetilde{\mathcal{F}}^T_{u \trianglelefteq e} \widetilde{\mathcal{F}}_{u \trianglelefteq e}.$$

Equation 4 is obtained by combining this equation with the previous one and factoring each summation by $(\widetilde{\mathcal{F}}_{u \trianglelefteq e})^T$.

## D  PROOFS OF OUR THEOREMS

**Theorem 1.** *$L^{\widetilde{\mathcal{F}}}$ is Hermitian and $L^{\widetilde{\mathcal{F}}} \succeq 0$, and the same holds for $L^{\widetilde{\mathcal{F}}}_N$.*

*Proof.* By definition, we have $L^{\widetilde{\mathcal{F}}} := \widetilde{\delta}^* \widetilde{\delta}$. Therefore, for any pair of indices $u, v \in V$, $L^{\widetilde{\mathcal{F}}}_{uv} = \widetilde{\delta}^*_{\bullet u} \widetilde{\delta}_{\bullet v}$ and $L^{\widetilde{\mathcal{F}}}_{vu} = \widetilde{\delta}^*_{\bullet v} \widetilde{\delta}_{\bullet u}$. Since this implies $L^{\widetilde{\mathcal{F}}}_{uv} = (L^{\widetilde{\mathcal{F}}}_{vu})^*$ for all of its entries $u, v$, we conclude that the

matrix is Hermitian. It follows that the spectrum of $L^{\widetilde{\mathcal{F}}}$ is real. By following essentially the same arguments using $\widetilde{\delta}' := \widetilde{\delta} D^{-\frac{1}{2}}$, one can show that the spectrum of $L_N^{\widetilde{\mathcal{F}}}$ is real as well.

Since, again by definition, $L^{\widetilde{\mathcal{F}}} = \widetilde{\delta}^* \widetilde{\delta}$, its associated quadratic form $x^* \widetilde{\delta}^* \widetilde{\delta} x$ (with $x \in \mathbb{C}$) can be rewritten as $x^* \widetilde{\delta}^* \widetilde{\delta} x = (\widetilde{\delta} x)^* (\widetilde{\delta} x) = ||\widetilde{\delta} x||_2^2$. Since $||\widetilde{\delta} x||_2^2$ is a norm, $||\widetilde{\delta} x||_2^2 \geq 0$ holds for all $x \in \mathbb{C}$, thus implying $L^{\widetilde{\mathcal{F}}} \succeq 0$ for all $x^* \in \mathbb{C}$. Thus, $L^{\widetilde{\mathcal{F}}} \succeq 0$. By following the same arguments using $\widetilde{\delta}' := \widetilde{\delta} D^{-\frac{1}{2}}$, one can show that $L_N^{\widetilde{\mathcal{F}}} \succeq 0$ as well. $\qquad\square$

**Theorem 2.** $L^{\widetilde{\mathcal{F}}_N} \preceq 2I.$

*Proof.* Let $Q_N^{\widetilde{\mathcal{F}}} := D^{-\frac{1}{2}} \widetilde{\delta}^* \widetilde{\delta} D^{-\frac{1}{2}}$ for the case where $\widetilde{\delta}$ has *not* been given an arbitrary orientation (this is in line with the classical construction of the Signless Laplacian $Q$ for undirected unweighted graphs). Since $Q_N^{\widetilde{\mathcal{F}}}$ is the product of a matrix and its conjugate, we have $Q_N^{\widetilde{\mathcal{F}}} \succeq 0$. It is easy to show that $Q_N^{\widetilde{\mathcal{F}}} = 2I - \hat{L}_N^{\mathcal{F}}$. From this, we deduce:

$$Q_N^{\widetilde{\mathcal{F}}} = 2I - L_N^{\widetilde{\mathcal{F}}} \succeq 0 \Leftrightarrow -L_N^{\widetilde{\mathcal{F}}} \succeq -2I \Leftrightarrow L_N^{\widetilde{\mathcal{F}}} \preceq 2I.$$

This shows that not only $L_N^{\widetilde{\mathcal{F}}}$ has a nonnegative spectrum, but also that its spectrum is upper-bounded by 2. $\qquad\square$

**Theorem 3.** *If $G$ is undirected, $L^{\widetilde{\mathcal{F}}}$ coincides with the classical sheaf Laplacian $L^{\mathcal{F}}$ for any choice of $q \in \mathbb{R}$. Also, if the sheaf is trivial and $G$ is undirected and unweighted, $L^{\widetilde{\mathcal{F}}}$ coincides with the classical graph Laplacian $L$. If $G$ is directed and we set $q = 0$, $L^{\widetilde{\mathcal{F}}}$ coincides with the classical sheaf Laplacian associated with the undirected version of $G$.*

*Proof.* **Part 1.** If $G$ is undirected, all restriction maps of the Directed Cellular Sheaf are real for every choice of $q \in \mathbb{R}$–this is because, for all $u, v \in V$, $A = A^T$ implies $\Re(T_{uv}^{(q)}) = 1$ and $\Im(T_{uv}^{(q)}) = 0$ for any choice of $q$. This implies $\Im(\widetilde{\mathcal{F}}_{u \trianglelefteq e}) = 0$ for all $e \in E$ where $u$ is one of its endpoints; therefore, $L^{\widetilde{\mathcal{F}}}$ is real valued and $L^{\widetilde{\mathcal{F}}} = L^{\mathcal{F}}$.
**Part 2.** Under the same assumptions on $G$, if the Directed Cellular Sheaf is trivial, $d = 1$ and $\widetilde{\mathcal{F}}_{u \trianglelefteq e} = 1$ for all edges $e \in E$ with $u$ being one if its endpoints. Thus, $L_{uv}^{\widetilde{\mathcal{F}}} = -1$ if $\{u, v\} \in E$ and $0$ otherwise, while $L_{uu}^{\widetilde{\mathcal{F}}} = |\{e \in E : e = \{u, v\}\}|$; by definition, it follows that $L^{\widetilde{\mathcal{F}}}$ coincides with the classical Laplacian matrix $L = D - A$ with $A \in \{0, 1\}^{n \times n}$.
**Part 3.** Setting $q = 0$ leads to, for all $u, v \in V$, $T_{uv}^{(q)} = \cos(0) + \mathbf{i}\sin(0) = 1$. Thus, $L^{\widetilde{\mathcal{F}}}$ coincides with the Directed Sheaf Laplacian $L^{\mathcal{F}}$ associated with the undirected version of $G$ which is obtained from it by preserving each of its edges and making all of them undirected—this coincides with discarding $\Im(\widetilde{\mathcal{F}}_{u \trianglelefteq e}) = 0$ for all $e \in E$ where $u$ is one of its endpoints. $\qquad\square$

**Theorem 4.** *Letting $G$ be a directed graph with unit weights, the Directed Sheaf Laplacian $L^{\widetilde{\mathcal{F}}}$ associated with a Trivial Directed Cellular Sheaf coincides with the Magnetic Laplacian $L^{(q)}$. In the special case where $q = \frac{1}{4}$, $L^{\widetilde{\mathcal{F}}}$ also coincides with the Sign-Magnetic Laplacian $L^\sigma$.*

*Proof.* **Part 1.** First, we show that, when adopting a Trivial Directed Cellular Sheaf for a directed graph $G$ with unit weights, we have:

$$L_{uv}^{\widetilde{\mathcal{F}}} = -T_{uv}^{(q)} \qquad u, v \in V : u \neq v$$
$$L_{uu}^{\widetilde{\mathcal{F}}} = |\Gamma(u)| \qquad u \in V.$$

Eq. 2 and 3 read:

$$L_{uv}^{\widetilde{\mathcal{F}}} = \begin{cases} -\widetilde{\mathcal{F}}_{u \trianglelefteq e}^* \widetilde{\mathcal{F}}_{v \trianglelefteq e} = -\widetilde{\mathcal{F}}_{u \trianglelefteq e}^T \widetilde{\mathcal{F}}_{v \trianglelefteq e}^0 T_{uv}^{(q)} & \text{if } e = (u,v) \\ -\widetilde{\mathcal{F}}_{u \trianglelefteq e}^* \widetilde{\mathcal{F}}_{v \trianglelefteq e} = -(\widetilde{\mathcal{F}}_{u \trianglelefteq e}^0 T_{vu}^{(q)})^* \widetilde{\mathcal{F}}_{v \trianglelefteq e} & \text{if } e = (v,u) \\ -\widetilde{\mathcal{F}}_{u \trianglelefteq e}^* \widetilde{\mathcal{F}}_{v \trianglelefteq e} = -\widetilde{\mathcal{F}}_{u \trianglelefteq e}^T \widetilde{\mathcal{F}}_{v \trianglelefteq e} & \text{if } e = \{u,v\} \\ 0 & \text{otherwise} \end{cases}$$

$$L_{uu}^{\widetilde{\mathcal{F}}} = \sum_{e \in \Gamma(u)} \widetilde{\mathcal{F}}_{u \trianglelefteq e}^* \widetilde{\mathcal{F}}_{u \trianglelefteq e}.$$

When considering a Trivial Directed Cellular Sheaf, we have

- $\widetilde{\mathcal{F}}_{u \trianglelefteq e} = \widetilde{\mathcal{F}}_{v \trianglelefteq e} = 1$ if $e = \{u,v\} \in E$ and, thus, $L_{uv}^{\widetilde{\mathcal{F}}} = -1 = -T_{uv}^{(q)}$ (th latter is because $A_{uv} = A_{vu}$ implies $T_{uv}^{(q)} = \cos(0) + \mathbf{i}\sin(0) = 1$).

- $\widetilde{\mathcal{F}}_{u \trianglelefteq e} = 1$ and $\widetilde{\mathcal{F}}_{v \trianglelefteq e} = T_{uv}^{(q)}$ if $e = (u,v) \in E$ and, thus, $L_{uv}^{\widetilde{\mathcal{F}}} = -T_{uv}^{(q)}$;

- $\widetilde{\mathcal{F}}_{u \trianglelefteq e} = T_{vu}^{(q)}$ and $\widetilde{\mathcal{F}}_{v \trianglelefteq e} = 1$ if $e = (v,u) \in E$ and, thus, $L_{uv}^{\widetilde{\mathcal{F}}} = -(T_{vu}^{(q)})^* = -T_{uv}^{(q)}$.

Each diagonal term $L_{uu}^{\widetilde{\mathcal{F}}}$ of $L^{\widetilde{\mathcal{F}}}$ reads

$$L_{uu}^{\widetilde{\mathcal{F}}} = \sum_{e \in \Gamma(u)} \widetilde{\mathcal{F}}_{u \trianglelefteq e}^* \widetilde{\mathcal{F}}_{u \trianglelefteq e} = \sum_{e=(u,v) \in E} \underbrace{\widetilde{\mathcal{F}}_{u \trianglelefteq e}^T \widetilde{\mathcal{F}}_{u \trianglelefteq e}}_{=1}$$

$$+ \sum_{e=(v,u) \in E} \underbrace{(\widetilde{\mathcal{F}}_{u \trianglelefteq e}^0)^T (T_{vu}^{(q)})^* \widetilde{\mathcal{F}}_{u \trianglelefteq e}^0 T_{vu}^{(q)}}_{=(T_{vu}^{(q)})^*(T_{vu}^{(q)})=1} + \sum_{e=\{u,v\} \in E} \underbrace{\widetilde{\mathcal{F}}_{u \trianglelefteq e}^T \widetilde{\mathcal{F}}_{u \trianglelefteq e}}_{=1}$$

$$= |\Gamma(u)|,$$

where $(T_{vu}^{(q)})^*(T_{vu}^{(q)}) = 1$ holds since $T_{vu}^{(q)} = \mathbf{i}$. With this, Part 1 is shown.

**Part 2.**

The Magnetic Laplacian reads

$$L^{(q)} := D_s - H^{(q)}, \text{ with } H^{(q)} := A_s \odot \exp\left(\mathbf{i}\, 2\pi q \left(A - A^\top\right)\right),$$

with $A_s := \frac{A + A^T}{2}$ and $D_s = \mathrm{diag}(\mathbf{1}_n A_s)$.

By definition we gave of $T_{uv}^{(q)}$, for a component $u,v$ with $u,v \in V$, we have:

$$L_{uv}^{(q)} := D_{s_{uv}} - H_{uv}^{(q)} = D_{s_{uv}} - A_{s_{uv}} T_{uv}^{(q)}.$$

**Part 2a.** Let's assume $G$ undirected. In such a case, we have we have $A_{s_{uv}} = 1$ whenever $\{u,v\} \in E$ and $A_{s_{uv}} = 0$ otherwise. This implies $D_{s_{uu}} = |\Gamma(u)|$. Thus, we have:

$$L_{uv}^{(q)} = -A_{s_{uv}} T_{uv}^{(q)} = -T_{uv}^{(q)} = L_{uv}^{\widetilde{\mathcal{F}}} \qquad\qquad u,v \in V : u \neq v$$

$$L_{uu}^{(q)} = D_{s_{uu}} - A_{s_{uu}} T_{uu}^{(q)} = D_{s_{uu}} = |\Gamma(u)| = L_{uu}^{\widetilde{\mathcal{F}}} \qquad\qquad u \in V,$$

where the last equation holds since $T_{uu}^{(q)} = 0$ for any $q$. Thus, $L^{\widetilde{\mathcal{F}}} = L^{(q)}$.

**Part 2b.** Let's assume $G$ directed without digons. In such a case, we have $A_{s_{uv}} = \frac{1}{2}$ whenever either $(u,v) \in E$ or $(v,u) \in E$ and $A_{s_{uv}} = 0$ otherwise. This implies $D_{s_{uu}} = \frac{1}{2}|\Gamma(u)|$. Thus, we have:

$$L_{uv}^{(q)} = -A_{s_{uv}} T_{uv}^{(q)} = -\frac{1}{2} T_{uv}^{(q)} = \frac{1}{2} L_{uv}^{\widetilde{\mathcal{F}}} \qquad\qquad u,v \in V : u \neq V$$

$$L_{uu}^{(q)} = D_{s_{uu}} - A_{s_{uu}} T_{uu}^{(q)} = D_{s_{uu}} = \frac{1}{2}|\Gamma(u)| = \frac{1}{2} L_{uu}^{\widetilde{\mathcal{F}}} \qquad\qquad u \in V,$$

where the last equation holds since $T_{uu}^{(q)} = 0$ for any $q$. Thus, $L^{\widetilde{\mathcal{F}}} = 2L^{(q)}$. Notice that the scaling factor is immaterial when the Laplacian matrix is embedded in a GCN/SNN, as it is directly subsumed by either $W_1$ or $W_2$ in equation 8 (only by the latter in a GCN, where $W_1$ is not present).

**Part 3.** Since, as shown in Fiorini et al. (2023), $L^{(q)}$ and $L^{\sigma}$ coincide with $q = \frac{1}{4}$, the last part of the claim follows directly from Parts 2a and 2b. □

**Theorem 5.** *Let $G$ be a directed graph with unit weights. Assuming a Trivial Directed Cellular Sheaf, the conjugate transpose $\widetilde{\delta}^*$ of the Directed Coboundary Operator $\widetilde{\delta}$ boils down to the complex-valued node-to-edge incidence matrix $\widehat{B} \in \mathbb{C}^{n \times m}$ defined for an edge $e \in E$ incident to a vertex $u$:*

$$\widehat{B}_{ue} = \begin{cases} 1 & \text{if } e = (u, v) \text{ or } e = \{u, v\} \text{ with } u < v \\ -1 & \text{if } e = \{u, v\} \text{ with } u > v \\ -T_{uv}^{(q)} & \text{if } e = (v, u). \end{cases}$$

*It follows that $L^{(q)} = \widehat{B}\widehat{B}^*$. With $q = \frac{1}{4}$, $L^{(\frac{1}{4})} = L^{\sigma} = \widehat{B}\widehat{B}^*$.*

*Proof.* (First, notice the arbitrary orientation that was given to the undirected edges).

From the proof of the previous theorem, we know that, if $G$ has unit weights and the Directed Cellular Sheaf is trivial, we have:

$$L_{uv}^{\widetilde{\mathcal{F}}} = -T_{uv}^{(q)} \qquad u, v \in V : u \neq v$$
$$L_{uu}^{\widetilde{\mathcal{F}}} = |\Gamma(u)| \qquad u \in V.$$

Let's consider $(\widehat{B}\widehat{B}^*)_{uv} = \sum_{e' \in E} \widehat{B}_{ue'}(\widehat{B}_{ve'})^*$. Since we are considering a graph, $u, v$ can only share a single edge. Calling it $e$, we have $(\widehat{B}\widehat{B}^*)_{uv} = \widehat{B}_{ue}(\widehat{B}_{ve})^*$ if $e \in E$ or 0 if they share no edge at all. Let's assume they do, and considering three cases:

- If $e = \{u, v\}$ with $u < v$, $\widehat{B}_{ue} = 1$ and $(\widehat{B}_{ve})^* = -1$ with an arbitrary orientation and, thus, $\widehat{B}_{ue}(\widehat{B}_{ve})^* = -1 = -T_{uv}^{(q)}$ (this is correct since, as shown before, $T_{uv}^{(q)}$ is always equal to 1 if $A_{uv} = A_{vu}$).

- If $e = \{u, v\}$ with $u > v$, $\widehat{B}_{ue} = -1$ and $(\widehat{B}_{ve})^* = 1$ with an arbitrary orientation and, thus, $\widehat{B}_{ue}(\widehat{B}_{ve})^* = -1 = -T_{uv}^{(q)}$ (as shown before, the latter is always equal to 1 if $A_{uv} = A_{vu}$).

- If $e = (u, v)$, $\widehat{B}_{ue} = 1$ and $(\widehat{B}_{ve})^* = (-T_{vu}^{(q)})^*$ and, thus, $\widehat{B}_{ue}(\widehat{B}_{ve})^* = (-T_{vu}^{(q)})^* = -T_{uv}^{(q)}$ since $T^{(q)}$ is Hermitian by construction.

- If $e = (v, u)$, $\widehat{B}_{ue} = -T_{uv}^{(q)}$ and $(\widehat{B}_{ve})^* = 1$ and, thus, $\widehat{B}_{ue}(\widehat{B}_{ve})^* = -T_{uv}^{(q)}$.

This shows that, if $G$ has unit weights and assuming a Trivial Directed Cellular Sheaf, we have $L^{\widetilde{\mathcal{F}}} = \widehat{B}\widehat{B}^*$. The fact that (with a scaling factor of 2, when needed) $L^{(q)} = \widehat{B}\widehat{B}^*$ and $L^{\sigma} = \widehat{B}\widehat{B}^*$ when $q = \frac{1}{4}$ follow from the previous theorem. □

## E  COMPLEXITY OF DSNN

As mentioned in the paper, the complexity of DSNN coincides, asymptotically, with that of NSD. This is because the adoption of complex-valued restriction maps—which are specific to DSNN and not present in NSD—does not affect the asymptotic inference complexity of DSNN. This is because complex-valued synaptic weights, pre-activations, and activations only incur a constant multiplicative overhead (approximately a factor of 4) in the forward pass and, thus, do not alter the asymptotic complexity from the real-valued case analysis. To better see this, consider three complex-valued matrices:

$$A = A_R + \mathbf{i}A_I, \quad X = X_R + \mathbf{i}X_I, \quad Y = Y_R + \mathbf{i}Y_I,$$

with

$$A_R, A_I \in \mathbb{R}^{m \times n}, \quad X_R, X_I \in \mathbb{R}^{n \times p}, \quad Y_R, Y_I \in \mathbb{R}^{m \times p},$$

satisfying the complex linear equation $Y = AX$. This equation can be rewritten purely in the real domain using the *lifting* transformation:

$$X_{\mathbb{R}} = \begin{bmatrix} X_R \\ X_I \end{bmatrix} \in \mathbb{R}^{2n \times p}, \quad Y_{\mathbb{R}} = \begin{bmatrix} Y_R \\ Y_I \end{bmatrix} \in \mathbb{R}^{2m \times p}, \quad A_{\mathbb{R}} = \begin{bmatrix} A_R & -A_I \\ A_I & A_R \end{bmatrix} \in \mathbb{R}^{2m \times 2n},$$

so that $Y_{\mathbb{R}} = A_{\mathbb{R}} X_{\mathbb{R}}$ holds. Hence, complex-valued operations can be reduced to real-valued operations with a constant factor overhead, which is immaterial in the asymptotic complexity.

**Experimental Scalability**   We evaluate the scalability of DSNN relative to NSD by reporting computation time and memory usage in the node classification task. For DSNN, we report results using the most resource-demanding configuration (General), with $d = 4$, 16 hidden channels, and 2 layers. We obtained the results included in the Tables 4 and 5 using a single Nvidia RTX A6000 GPU.

## F  FURTHER DETAILS ON THE DATASETS

**Real-world dataset.**   The `Texas`, `Wisconsin`, and `Cornell` datasets are part of the WebKB collection, modeling links between websites from different universities. In these datasets, nodes are labeled as student, project, course, staff, or faculty.

The `Film` dataset is derived from a film–director–actor–writer network. Each node represents an actor, and edges indicate co-occurrence on the same Wikipedia page. Node features correspond to keywords extracted from these Wikipedia pages. The nodes are classified into five categories based on the content of the actors' Wikipedia entries.

The `Citeseer` dataset contains 3,312 scientific publications classified into six categories. The citation network includes 4,732 links. Each publication is represented by a binary word vector indicating the presence or absence of words from a dictionary of 3,703 unique terms.

The `PubMed` dataset consists of 19,717 scientific publications related to diabetes, categorized into three classes. The citation network contains 44,338 links. Each publication is described by a TF-IDF weighted word vector derived from a dictionary of 500 unique words.

The `Cora` dataset includes 2,708 scientific publications classified into seven classes, with a citation network comprising 5,429 links. Each publication is represented by a binary word vector indicating the presence or absence of words from a dictionary of 1,433 unique terms.

The `Squirrel` and `Chameleon` datasets consist of articles from the English Wikipedia (December 2018). Nodes represent articles, and edges represent mutual links between them. Node features indicate the presence of specific nouns in the articles. Nodes are grouped into five categories based on the original regression targets.

`Telegram` is an influence network that analyses the interactions and influences between distinct groups and actors who associate and propagate political ideologies. This is a pairwise-influence network between 245 Telegram channels with 8912 links. The labels are generated following the method discussed in (Bovet & Grindrod, 2020), with a total of four classes.

The `Questions` dataset is derived from the question-answering platform Yandex Q. Nodes represent users, and an edge connects two nodes if one user answered the other's question within a one-year interval (September 2021–August 2022). To limit dataset size, Platonov et al. (2023) focus on users interested in the topic "medicine." The task is to predict which users remained active (i.e., were not deleted or blocked) by the end of the period. Node features are computed as the mean of fastText embeddings (Grave et al., 2018) of words in the user description, with an additional binary feature indicating the 15% of users lacking a description. The final dataset contains 48.9K nodes with an average degree of 6.28.

Table 7 reports the statistics of the datasets used in this paper.

**Synthetic dataset.**   Following Zhang et al. (2021b), we generate synthetic graphs using the directed stochastic block model (DSBM) as follows. Let $n$ be the number of nodes and $C$ the number of equal-sized communities $\{C_1, \ldots, C_C\}$. First, we sample an undirected graph by connecting each

Table 7: Summary of datasets: number of nodes, edges, density (%), and percentage of directed edges.

| Dataset | # Nodes | # Edges | Density (%) | % Directed Edges |
|---|---|---|---|---|
| Roman-Empire | 22,662 | 32,927 | 0.006 | 65.24 |
| Texas | 183 | 295 | 0.009 | 89.25 |
| Wisconsin | 251 | 466 | 0.007 | 89.11 |
| Film | 7,600 | 26,752 | 0.046 | 87.74 |
| Squirrel | 2,223 | 46,998 | 0.009 | 90.60 |
| Chameleon | 890 | 8,854 | 1.120 | 85.01 |
| Cornell | 183 | 280 | 0.008 | 93.50 |
| Telegram | 245 | 8,912 | 14.900 | 90.33 |
| Citeseer | 3,327 | 4,676 | 0.042 | 98.78 |
| PubMed | 18,717 | 44,327 | 0.013 | 99.97 |
| Cora | 2,708 | 5,278 | 0.072 | 97.14 |
| Questions | 48,921 | 153,540 | 0.064 | 98.97 |

pair of nodes $u \in C_i$ and $v \in C_j$ independently with probability $\alpha_{ij} \in [0, 1]$, $\alpha_{ij} = \alpha_{ji}$, where $\alpha_{ii}$ controls intra-community edge density and $\alpha_{ij}$ for $i \neq j$ controls inter-community connectivity. To obtain a directed graph, we introduce a rule to transform the graph from undirected to directed: we define a collection of probabilities $\{\beta_{ij}\}_{1 \leq i,j \leq C}$, where $\beta_{ij} \in [0, 1]$, such that $\beta_{ij} + \beta_{ji} = 1$. If $u \in C_i$ and $v \in C_j$, we orient the edge $u \to v$ with probability $\beta_{ij}$, and $v \to u$ with probability $\beta_{ji}$.

The synthetic datasets we used do not exhibit strong homophily. On the contrary, they lack significant homophilic structure, as shown by the following table, where their homophily is measured according to both (Pei et al., 2020) and (Lim et al., 2021):

| Dataset | Homophily (Pei et al., 2020) | Homophily (Lim et al., 2021) |
|---|---|---|
| Synthetic with $\alpha_{ij} = 0.05$ | 0.33 | 0.36 |
| Synthetic with $\alpha_{ij} = 0.08$ | 0.23 | 0.25 |
| Synthetic with $\alpha_{ij} = 0.10$ | 0.20 | 0.20 |

Table 8: Homophily values for synthetic datasets under different $\alpha_{ij}$ settings.

## G  FURTHER DETAILS ON THE EXPERIMENTS

**Hardware.**  The experiments were conducted on 2 different machines:

1. An Intel(R) Xeon(R) Gold 6326 CPU @ 2.90GHz with 380 GB RAM, equipped with an NVIDIA Ampere A100 40GB.

2. A 12th Gen Intel(R) Core(TM) i9-12900KF CPU @ 3.20GHz CPU with 64 GB RAM, equipped with an NVIDIA RTX 4090 GPU.

3. A Intel(R) Xeon(R) Silver 4210R CPU @ 2.40GHz with 64 GB RAM, equipped with an NVIDIA A100 80GB.

4. A AMD Ryzen Threadripper PRO 5975WX 32-Cores with 256 GB RAM, equipped with an NVIDIA RTX A6000 48 GB.

**Model Settings.**  We trained every learning model considered in this paper for up to 1000 epochs with early stops of 200. We adopted a learning rate of $\{1 \cdot 10^{-2}, 2 \cdot 10^{-2}, 5 \cdot 10^{-3}\}$ and employed the optimization algorithm Adam.

We adopted a hyperparameter optimization procedure to identify the best set of parameters for each model. For every model, we searched for the optimal combination of the following hyperparameters for the link prediction:

- **Dropout:** {0.0, 0.1, 0.2, 0.3, 0.4, 0.5, 0.6, 0.7, 0.8, 0.9}

- **Number of layers:** {2, 3, 4, 5, 6}

- **Hidden channels:** {8, 16, 32, 64}.

For some specific models, we also included additional hyperparameters in the search space:

- **NSD-comp and NSD:** `sheaf_act` $\in$ {`elu`, `tanh`, `relu`}; $d \in \{2, 3, 4, 5\}$; `add_lp` $\in$ {`True`, `False`}; `add_hp` $\in$ {`True`, `False`}

- **DirGNN:** $\alpha_{\text{DirGNN}} \in \{0.0, 0.1, 0.2, 0.3, 0.4, 0.5, 0.6, 0.7, 0.8, 0.9, 1.0\}$; `jk` $\in$ {`cat`, `max`}

- **MagNet:** $q \in \{0.0, 0.05, 0.1, 0.15, 0.2, 0.25\}$

- **GCNII:** $\alpha_{\text{GCNII}} \in \{0.0, 0.1, 0.2\}$; $\lambda \in \{0.0, 1.0, 1.5\}$

- **FAGCN:** $\varepsilon \in \{0.0, 0.1, 0.2, 0.3, 0.4, 0.5, 0.6, 0.7, 0.8, 0.9, 1.0\}$

- **GGCN:** `decay_rate` $\in$ {0.0, 0.1, 0.2, 0.3, 0.4, 0.5, 0.6, 0.7, 0.8, 0.9, 1.0, 1.1, 1.2}

- **GPRGNN:** $\alpha_{\text{GPRGNN}} \in \{0.0, 0.1, 0.2, 0.3, 0.4, 0.5, 0.6, 0.7, 0.8, 0.9, 1.0\}$; `dprate_GPRGNN` $\in$ {0.0, 0.1, 0.2, 0.3, 0.4, 0.5, 0.6, 0.7, 0.8, 0.9}

- **DSNN:** $q \in \{0.0, 0.05, 0.1, 0.15, 0.2, 0.25\}$, `sheaf_act` $\in$ {`elu`, `tanh`, `relu`}, $d \in \{2, 3, 4, 5\}$; `add_lp` $\in$ {`True`, `False`}; `add_hp` $\in$ {`True`, `False`}

## H  SENSITIVITY ANALYSIS ON $q$

The values of $q$ for DSNN, alongside those reported by MagNet (Zhang et al., 2021b), are summarized in Table 9. The results indicate that both approaches tend to favor either small (close to 0) or large (close to 0.25) values for $q$. This analysis suggests that there is no clear correlation between heterophily/homophily and the use of directionality in either DSNN or MagNet. Instead, both models exploit edge directionality adaptively, using it when it enhances information propagation for the learning task at hand. In particular, the table shows that, in 30 out of 36 computations involving DSNN, setting a strictly positive $q$ leads to a better performance.

Table 9: Best $q$ values used for DSNN variants compared to MagNet (Zhang et al., 2021b) across different datasets

| | Roman-Empire | Texas | Wisconsin | Film | Squirrel | Chameleon | Cornell | Telegram | Citeseer | Pubmed | Cora | Questions |
|---|---|---|---|---|---|---|---|---|---|---|---|---|
| **Diag-DSNN** | 0.1 | 0 | 0.1 | 0.1 | 0.25 | 0.1 | 0.25 | 0.1 | 0.25 | 0.2 | 0.25 | 0.2 |
| **O(d)-DSNN** | 0.1 | 0 | 0.2 | 0.2 | 0.25 | 0 | 0 | 0.1 | 0 | 0.2 | 0.25 | 0.15 |
| **Gen-DSNN** | 0.15 | 0.1 | 0.2 | 0.2 | 0.15 | 0.1 | 0 | 0.2 | 0.25 | 0.25 | 0.1 | 0.05 |
| MagNet | 0.20 | 0.15 | 0 | 0.1 | 0.15 | 0.05 | 0.15 | 0.15 | 0.15 | 0.2 | 0 | 0.25 |

Table 10 reports the best $q$ values for the direction prediction task. In 27 out of 30 cases, incorporating directionality (i.e., setting $q \neq 0$) improves performance, further confirming the benefits of explicitly modeling edge orientation.

Table 10: Best $q$ values for the direction prediction task. Incorporating directionality improves performance in most cases

| $q$ | Texas | Wisconsin | Film | Squirrel | Chameleon | Cornell | Citeseer | Pubmed | Cora | Questions |
|---|---|---|---|---|---|---|---|---|---|---|
| **Diag-DSNN** | 0.25 | 0.25 | 0.05 | 0.25 | 0.05 | 0 | 0.05 | 0.25 | 0.15 | 0.15 |
| **O(d)-DSNN** | 0.25 | 0 | 0.1 | 0.1 | 0.2 | 0.10 | 0.20 | 0.25 | 0.05 | 0.05 |
| **Gen-DSNN** | 0 | 0.25 | 0.15 | 0.1 | 0.2 | 0.10 | 0.10 | 0.2 | 0.05 | 0.25 |
| MagNet | 0.20 | 0.10 | 0.10 | 0.15 | 0.10 | 0.25 | 0.15 | 0.10 | 0.10 | 0 |

Table 11 reports the results of a sensitivity analysis on $q$ carried out on the synthetic datasets. The table confirms that the best performance is obtained with $q \neq 0$ also on this dataset.

Table 11: Sensitivity analysis on $q$ for synthetic datasets. Each entry reports mean and standard deviation accuracy

| $\alpha/q$ | 0.00 | 0.10 | 0.20 | 0.25 |
|---|---|---|---|---|
| 0.05 | 20.40±1.05 | 98.00±0.93 | 98.16±0.77 | 97.28±0.78 |
| 0.08 | 19.64±1.03 | 97.10±0.70 | 96.74±0.77 | 88.78±1.21 |
| 0.10 | 20.18±0.76 | 98.60±0.46 | 99.14±0.36 | 98.66±0.35 |
| Avg | 20.07±0.95 | 97.90±0.70 | 98.01±0.64 | 94.91±0.78 |

Table 12: Best $d$ values for each dataset.

| Model | Texas | Wisconsin | Film | Squirrel | Chameleon | Cornell | Citeseer | Pubmed | Cora | Telegram | Questions | Roman-Empire |
|---|---|---|---|---|---|---|---|---|---|---|---|---|
| Diag | 4 | 3 | 4 | 5 | 5 | 3 | 5 | 3 | 2 | 4 | 5 | 4 |
| Bundle | 4 | 5 | 4 | 5 | 5 | 3 | 5 | 5 | 4 | 3 | 3 | 3 |
| General | 3 | 5 | 4 | 5 | 4 | 4 | 3 | 3 | 3 | 4 | 3 | 3 |

# I  SENSITIVITY ANALYSIS ON $d$

Table 12 reports the best $d$ values selected for each dataset and model variant. While the optimal $d$ varies across datasets and models, some clear patterns emerge. The Diag model tends to favor smaller values on smaller datasets (e.g., Wisconsin, Cora), whereas Bundle and General models often require slightly higher $d$ on larger or more complex graphs (e.g., Squirrel, Chameleon). Overall, the most common best $d$ value across all datasets and models is 4, the minimum value of 2 never occurs, and the selected $d$ generally ranges from 3 to 5. This demonstrates that moderate to slightly higher $d$ values are consistently preferred, and that $d$ should be adapted based on both the model variant and dataset characteristics to achieve optimal performance.

We also perform the scalability of the method by selecting different $d$ and calculate the peak GPU memory (MiB). We adopt the same setup as reported in Appendix E, corresponding to the most resource-demanding configuration (General) with 16 hidden channels, and 2 layers.

Table 13: Peak GPU usage (MiB) for different values of $d$.

| $d$ | Texas | Cornell | Wisconsin | Cora | Citeseer | Pubmed | Film | Squirrel | Chameleon | Questions | Roman-Empire |
|---|---|---|---|---|---|---|---|---|---|---|---|
| 2 | 402 | 402 | 406 | 598 | 823 | 1453 | 926 | 1103 | 548 | 3514 | 1414 |
| 3 | 403 | 403 | 407 | 600 | 826 | 1573 | 981 | 1207 | 565 | 3904 | 1516 |
| 4 | 404 | 404 | 408 | 606 | 830 | 1744 | 1078 | 1362 | 598 | 4462 | 1646 |
| 5 | 404 | 404 | 409 | 630 | 834 | 1954 | 1204 | 1567 | 636 | 5173 | 1812 |
| 10 | 417 | 417 | 428 | 842 | 952 | 3667 | 2203 | 3253 | 960 | 10966 | 3135 |
| 15 | 437 | 437 | 458 | 1183 | 1253 | 6469 | 3850 | 6058 | 1497 | 20500 | 5295 |

Table 13 reports GPU memory usage (in MiB) across all datasets for different values of $d$. Even for the largest configuration considered ($d = 15$), our method runs on a 24GB GPU for small and medium datasets (e.g., Texas, Cornell, Wisconsin), demonstrating its efficiency and scalability. For larger datasets such as Pubmed, Questions, and Squirrel, memory usage increases with d as expected, yet remains fully manageable, showing that our approach can handle both small and large-scale graphs without exceeding typical hardware limits.

# J  EXPLICIT CONSTRUCTION AND NUMERIC EXAMPLE OF $L^{\widetilde{\mathcal{F}}}$ FOR $q = 0.25$

## J.1  SETUP AND DEFINITIONS

Consider the graph $G = (V, E)$ with node set $V = \{A, B, C\}$. We equip $G$ with a directed cellular sheaf $\widetilde{\mathcal{F}}$ of stalk dimension $d = 2$. The directionality parameter is $q \in \mathbb{R} \setminus \{0\}$ and we define $\theta := 2\pi q$.

**Edges and Adjacency.**

- $e_1$: a directed arc $A \to B$. Adjacency: $A_{AB} = 1$, $A_{BA} = 0$.
- $e_2, e_3$: an undirected connection $B - C$, modeled as two antiparallel directed arcs $e_2 : B \to C$ and $e_3 : C \to B$. Adjacency: $A_{BC} = A_{CB} = 1$.

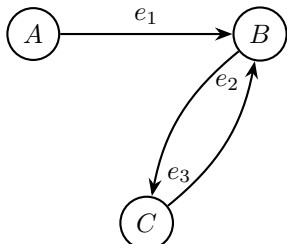

Figure 2: Visualization of the graph $G = (V, E)$ with nodes $V = A, B, C$. Edge $e_1$ represents the directed arc $A \to B$, while edges $e_2$ and $e_3$ form two antiparallel arcs between $B$ and $C$, modeling an undirected connection.

**Magnetic Restriction Maps.** For an edge $e = (u, v)$, the restriction map at the target node $v$ is modulated by a magnetic phase:

$$\mathcal{F}_{v \trianglelefteq e} = \mathcal{F}^0_{v \trianglelefteq e} T^{(q)}_{uv}, \quad \text{with } T^{(q)}_{uv} = \exp(\mathbf{i}\theta(A_{uv} - A_{vu})) I_d,$$

and we assume the source restriction maps $\mathcal{F}_{u \trianglelefteq e}$ are the identity $I_d$.

**Phase Factor Calculation.**

- Edge $A \to B$: $A_{AB} - A_{BA} = 1$, phase $\tau = \exp(\mathbf{i}\theta) = \cos(\theta) + \mathbf{i}\sin(\theta)$.
- Edge $B - C$: $A_{BC} - A_{CB} = 0$, phase $1$.

**Matrix Assignments.** We define the base maps $\mathcal{F}^0_{v \trianglelefteq e}$ using $I_2$ and two matrices $M_1, M_2 \in \mathbb{R}^{2 \times 2}$:

- Edge $e_1$: $\mathcal{F}_{A \trianglelefteq e_1} = I_2$, $\mathcal{F}_{B \trianglelefteq e_1} = \tau I_2$.
- Edge $e_2$: $\mathcal{F}_{B \trianglelefteq e_2} = M_1$, $\mathcal{F}_{C \trianglelefteq e_2} = I_2$.
- Edge $e_3$: $\mathcal{F}_{C \trianglelefteq e_3} = M_2$, $\mathcal{F}_{B \trianglelefteq e_3} = I_2$.

## J.2 Matrix Construction

**Directed Coboundary Operator $\widetilde{\delta}$.** The operator $\widetilde{\delta} \in \mathbb{C}^{|E|d \times |V|d}$ acts on a cochain $x$ as $(\widetilde{\delta}x)_e = \mathcal{F}_{u \trianglelefteq e}x_u - \mathcal{F}_{v \trianglelefteq e}x_v$. Explicitly:

$$
\widetilde{\delta} = \begin{array}{c} \\ e_1 \\ e_2 \\ e_3 \end{array} \begin{array}{ccc} A & B & C \\ \left(\begin{matrix} I_2 & -\tau I_2 & 0_2 \\ 0_2 & M_1 & -I_2 \\ 0_2 & -I_2 & M_2 \end{matrix}\right) \end{array} = \begin{pmatrix} I_2 & -\tau I_2 & 0_2 \\ 0_2 & M_1 & -I_2 \\ 0_2 & -I_2 & M_2 \end{pmatrix}.
$$

**Directed Sheaf Laplacian $L^{\widetilde{\mathcal{F}}}$.** The Laplacian is defined as $L^{\widetilde{\mathcal{F}}} = \widetilde{\delta}^* \widetilde{\delta}$, with $\widetilde{\delta}^*$ the conjugate transpose:

$$
L^{\widetilde{\mathcal{F}}} = \begin{pmatrix} I_2 & 0_2 & 0_2 \\ -\bar{\tau} I_2 & M_1^\top & -I_2 \\ 0_2 & -I_2 & M_2^\top \end{pmatrix} \begin{pmatrix} I_2 & -\tau I_2 & 0_2 \\ 0_2 & M_1 & -I_2 \\ 0_2 & -I_2 & M_2 \end{pmatrix}.
$$

**Block Computation.** The block matrices are:

- $L^{\widetilde{\mathcal{F}}}_{AA} = I_2$, $L^{\widetilde{\mathcal{F}}}_{AB} = -\tau I_2$, $L^{\widetilde{\mathcal{F}}}_{BA} = -\bar{\tau} I_2$.
- $L^{\widetilde{\mathcal{F}}}_{BB} = 2I_2 + M_1^\top M_1$, $L^{\widetilde{\mathcal{F}}}_{BC} = -(M_1^\top + M_2)$, $L^{\widetilde{\mathcal{F}}}_{CB} = -(M_1 + M_2^\top)$.
- $L^{\widetilde{\mathcal{F}}}_{CC} = I_2 + M_2^\top M_2$.

**Explicit Matrix Decomposition.** With $\tau = \cos(\theta) + \mathbf{i}\sin(\theta)$ and $M_1 = \left(\begin{smallmatrix}1 & 1\\ 0 & 1\end{smallmatrix}\right)$, $M_2 = \left(\begin{smallmatrix}1 & 0\\ 1 & 1\end{smallmatrix}\right)$, we have

$$
L^{\widetilde{\mathcal{F}}} = \underbrace{\begin{pmatrix}
1 & 0 & -\cos(\theta) & 0 & 0 & 0\\
0 & 1 & 0 & -\cos(\theta) & 0 & 0\\
-\cos(\theta) & 0 & 3 & 1 & -2 & 0\\
0 & -\cos(\theta) & 1 & 4 & -2 & -2\\
0 & 0 & -2 & -2 & 3 & 1\\
0 & 0 & 0 & -2 & 1 & 2
\end{pmatrix}}_{\textbf{Real}} + \mathbf{i}\underbrace{\begin{pmatrix}
0 & 0 & -\sin(\theta) & 0 & 0 & 0\\
0 & 0 & 0 & -\sin(\theta) & 0 & 0\\
\sin(\theta) & 0 & 0 & 0 & 0 & 0\\
0 & \sin(\theta) & 0 & 0 & 0 & 0\\
0 & 0 & 0 & 0 & 0 & 0\\
0 & 0 & 0 & 0 & 0 & 0
\end{pmatrix}}_{\textbf{Imaginary}}
$$

This decomposition clearly shows that the imaginary part encodes edge directionality, while the real part encodes the undirected topology.

Setting $q = 0.25$, the magnetic phase factor becomes $\tau = \cos(2\pi q) + \mathbf{i}\sin(2\pi q) = \mathbf{i}$. Substituting this value into the Laplacian yields the explicit numeric form of the Directed Sheaf Laplacian:

$$
L^{\widetilde{\mathcal{F}}} = \underbrace{\begin{pmatrix}
1 & 0 & 0 & 0 & 0 & 0\\
0 & 1 & 0 & 0 & 0 & 0\\
0 & 0 & 3 & 1 & -2 & 0\\
0 & 0 & 1 & 4 & -2 & -2\\
0 & 0 & -2 & -2 & 3 & 1\\
0 & 0 & 0 & -2 & 1 & 2
\end{pmatrix}}_{\textbf{Real}} + \mathbf{i}\underbrace{\begin{pmatrix}
0 & 0 & -1 & 0 & 0 & 0\\
0 & 0 & 0 & -1 & 0 & 0\\
1 & 0 & 0 & 0 & 0 & 0\\
0 & 1 & 0 & 0 & 0 & 0\\
0 & 0 & 0 & 0 & 0 & 0\\
0 & 0 & 0 & 0 & 0 & 0
\end{pmatrix}}_{\textbf{Imaginary}}
$$

The **real component** encodes the undirected topology, wile the **imaginary component** captures all directional information introduced by the magnetic phase. This numeric example provides an explicit demonstration of how the Directed Sheaf Laplacian $L^{\widetilde{\mathcal{F}}}$ separates directional and undirectional contributions in a simple 3-node graph.

## K    SPECTRAL COMPARISON

Figure 3 depicts the spectra of the DSNN's Directed Sheaf Laplacian (with $d = 1$) and of the Magnetic Laplacian on Cornell, Texas, and Cora, for different values of the charge parameter $q$. The figure shows that there are no substantial differences between the two spectra when the stalk has dimension 1.

## L    LEARNABLE Q

While the parameter $q$ was treated as a fixed hyperparameter in the main paper, our architecture is compatible with learning $q$ jointly with the model. We trained Gen-DSNN with $q$ as a learnable scalar using three different initialization values (0.25, 0.125, 0). Table 1 reports the mean and standard deviation of the learned $q$ over 10 folds for five datasets, together with the fixed $q$ value that produced the best performance in the paper.

Table 14: Learned $q$ values (mean $\pm$ std)

| Dataset | init $q = 0.25$ | init $q = 0.125$ | init $q = 0$ | $q$ for best results |
|---|---|---|---|---|
| Cora | $0 \pm 0.008$ | $0 \pm 0.005$ | $0 \pm 0.004$ | 0.10 |
| Wisconsin | $0.15 \pm 0.06$ | $0.11 \pm 0.03$ | $0.05 \pm 0.06$ | 0.20 |
| Cornell | $0.18 \pm 0.08$ | $0.10 \pm 0.09$ | $0.01 \pm 0.02$ | 0.00 |
| Texas | $0.15 \pm 0.04$ | $0.14 \pm 0.06$ | $0.09 \pm 0.08$ | 0.10 |
| Telegram | $0.19 \pm 0.07$ | $0.10 \pm 0.09$ | $0.04 \pm 0.05$ | 0.20 |

We observe that the learned q converges stably across different initializations: for some datasets (e.g., Cora) it consistently converges close to zero, suggesting that directional information is less relevant, while for others (e.g., Wisconsin, Texas, Telegram) it converges to moderate values that are close to the best fixed choice used in the paper. This indicates that learning $q$ is both feasible and meaningful.

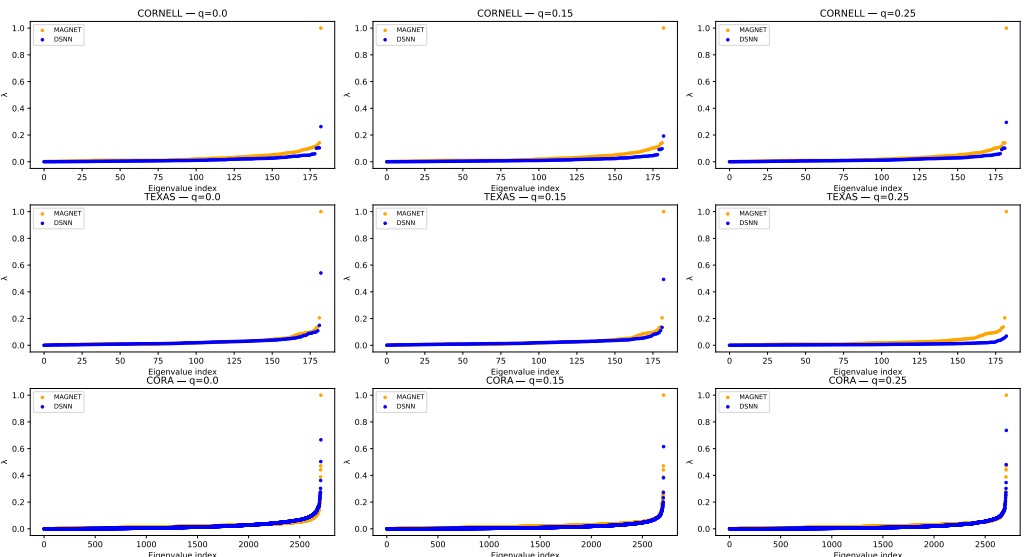

Figure 3: Spectra of the DSNN's Directed Sheaf Laplacian and of the Magnetic Laplacian on Cornell, Texas, and Cora, for different values of the charge parameter $q$ and $d = 1$.

