# OpenReview forum: "Sheaves Reloaded: A Direction Awakening"
_ICLR.cc/2026/Conference — ICLR 2026 Poster_

### Official Review · Reviewer_uTjh · 2025-10-26

**Soundness:** 3
**Presentation:** 3
**Contribution:** 2
**Rating:** 6
**Confidence:** 2

**Summary:**

This paper extends Sheaf Neural Networks (SNNs) to directed graphs by introducing the Directed Cellular Sheaf and the corresponding Directed Sheaf Laplacian (DSL), which explicitly encode edge orientation through complex-valued, direction-aware restriction maps. Building on this framework, the authors propose the Directed Sheaf Neural Network (DSNN), enabling principled learning on directed and heterophilic graphs. Experiments on synthetic and real-world datasets demonstrate that DSNN effectively captures directional dependencies and outperforms existing SNN and GNN models where edge directionality is crucial.

**Strengths:**

1. This paper introduces a mathematically principled extension of SNNs to directed graphs through the Directed Cellular Sheaf and Directed Sheaf Laplacian.

2. It effectively captures asymmetric and directional relationships while maintaining robustness to heterophily.

3. The experiments demonstrate consistent performance gains on both synthetic and real-world directed graph datasets.

**Weaknesses:**

1.  Most compared models are from 2020–2022, with only one from 2024. The evaluation lacks more recent direction-aware or topology-based GNNs, which weakens the empirical evidence for DSNN’s claimed advantages.

2. The paper does not clearly justify why Sheaf Neural Networks are the right framework for addressing heterophily or extending to directed graphs. Given the recent shift toward graph foundation models and unified architectures, it remains unclear whether adapting SNNs is the most effective or timely direction, rather than developing more generalizable approaches.

**Questions:**

1. The introduction should better articulate why extending SNNs remains valuable in 2025. Specifically, it should discuss the advantages of combining SNNs with GNNs for handling directed graphs and heterophily, and explain why these structural refinements are still meaningful in the era of more unified and general graph learning paradigms.

2. The comparison set is mostly limited to models from 2020–2022, with only one 2024 method included. More recent GNNs addressing heterophily and oversmoothing (from 2024–2025) should be added to strengthen the empirical evaluation and demonstrate DSNN’s relevance against state-of-the-art models.

3. The paper does not include baselines from graph prompting, graph foundation, or pre-trained graph models, which have recently become dominant in node classification and general graph learning. Including such comparisons would position the work more clearly within the modern landscape of graph representation learning.

---

> ### Author Response · Authors · 2025-11-20
>
> Thanks for reading our paper and giving us your opinion. This and the next posts contain our point-to-point replies to your comments and questions. For clarity, we compounded commends and questions that address similar aspects of our paper.
>
> **W**: The paper does not clearly justify why Sheaf Neural Networks are the right framework for addressing heterophily or extending to directed graphs. Given the recent shift toward graph foundation models and unified architectures, it remains unclear whether adapting SNNs is the most effective or timely direction, rather than developing more generalizable approaches.
>
> **Q**: The introduction should better articulate why extending SNNs remains valuable in 2025. Specifically, it should discuss the advantages of combining SNNs with GNNs for handling directed graphs and heterophily, and explain why these structural refinements are still meaningful in the era of more unified and general graph learning paradigms.
>
> **Q**:  The paper does not include baselines from graph prompting, graph foundation, or pre-trained graph models, which have recently become dominant in node classification and general graph learning. Including such comparisons would position the work more clearly within the modern landscape of graph representation learning.
>
> Our choice to build on Sheaf Neural Networks is motivated by a property that distinguishes them from recent unified GNN architectures and graph foundation models: sheaves provide an explicit, mathematically grounded mechanism for encoding how information should flow across each edge within a diffusion process. This mechanism is strictly richer than message-passing and is precisely what enables SNNs to model heterophily and directional asymmetry in a principled way, where the restriction maps explicitly specify how information should be transported across each edge to maximize the performance on the downstream task, so similarity across neighbors is not forcibly imposed by the architecture. This is why prior work has shown that SNNs substantially outperform message-passing neural nets (MPNNs) on heterophilic tasks.
>
> Directed graphs introduce a second type of asymmetry: information from i to j should not in general be treated as equivalent to information from j to i. Our directed sheaf construction allows this asymmetry to be encoded directly in the operator, via the complex-valued phase in the directional maps. Existing unified or foundation-style GNNs lack such a structural handle; they recover directionality only implicitly through learned weights.
>
> Additionally, sheaves arise naturally in algebraic topology as the canonical objects for encoding local-to-global consistency on cell complexes, making them the mathematically principled tool for modeling heterogeneous and direction-dependent interactions on graphs.
>
>
> While graph foundation and pre-trained models are increasingly influential, they are architecturally orthogonal to our contribution: they rely a graph operator but do not offer a mechanism to design or generalize one. Our work provides a strictly more expressive operator that can, in principle, be plugged into such models as a drop-in replacement for standard Laplacians or diffusion kernels. For this reason, comparisons with prompting or foundation models could be misleading, because our contribution operates at a different level: it upgrades the geometric operator those models depend on rather than competing with the prompting or pre-training framework itself.

---

> ### Author Response · Authors · 2025-11-20
>
> **W**: Most compared models are from 2020–2022, with only one from 2024. The evaluation lacks more recent direction-aware or topology-based GNNs, which weakens the empirical evidence for DSNN’s claimed advantages.
> **Q**: The comparison set is mostly limited to models from 2020–2022, with only one 2024 method included. More recent GNNs addressing heterophily and oversmoothing (from 2024–2025) should be added to strengthen the empirical evaluation and demonstrate DSNN’s relevance against state-of-the-art models.
>
> We added four new baselines: DiGCL [1], HaarNet [2], CAGN [3], and H2SGNN [4]. The results for node classification and link prediction are reported in the following tables.
>
> Table 1: Node classification on real-world dataset (mean ± std)
> | Model | Roman-Empire | Texas | Wisconsin | Film | Squirrel | Chameleon | Cornell | Telegram | Citeseer | Pubmed | Cora | Questions |
> | :---: | :---: | :---: | :---: | :---: | :---: | :---: | :---: | :---: | :---: | :---: | :---: | :---: |
> | **Diag-DSNN** | 90.40±0.31 | **88.65±4.95** | **90.20±4.02** | 38.34±1.01 | **45.37±2.21** | 46.84±4.03 | **87.84±5.70** | 94.42±3.03 | 79.80±1.49 | *90.23±0.44* | **87.36±1.41** | *79.08±0.72* |
> | **O(d)-DSNN** | *92.08±0.24* | *87.57±4.04* | 89.80±3.82 | 37.37±0.98 | 44.54±2.26 | 45.36±3.29 | 87.30±7.26 | *94.62±2.24* | 77.28±1.63 | 90.05±0.55 | *87.30±1.62* | *79.24±0.68* |
> |**Gen-DSNN** | **92.08±0.36** | *87.57±5.43* | *89.22±3.31* | **38.40±0.75** | 45.34±1.69 | **47.16±3.54** | **87.84±6.86** | **94.81±2.28** | **79.88±1.21** | **90.17±0.44** | 87.58±0.72 | **79.55±0.67** |
> | HaarNet | 85.42±0.43 | 77.57±4.18 | 71.56±6.69 | *36.38±1.01* | *40.52±3.14* | *42.43±3.98* | 73.91±7.57 | 91.12±3.69 | 76.51±1.64 | 88.39±0.61 | 82.68±1.54 | 75.01±0.94 |
> | CAGN | OOM | 75.67±7.15 | 84.11±4.51 | 34.86±1.06 | 35.38±0.99 | 39.79±3.89 | 73.79±7.25 | 86.73±3.69 | 73.64±2.81 | OOM | 86.23±1.05 | OOM |
> | DiGCL | 52.71±0.32 | 57.56±5.15 | 65.50±4.23 | 29.38±0.73 | 38.90±1.78 | 41.71±2.20 | 62.16±5.12 | 80.57±2.25 | *77.42±0.14* | 80.97±0.70 | 76.12±1.04 | OOM |
> | H2SGNN | 69.59±0.45 | 72.70±8.83 | 78.23±5.22 | 36.75±1.33 | 37.09±1.21 | 41.14±3.60 | 74.05±5.94 | 62.69±3.95 | 77.17±1.36 | 86.94±0.42 | 82.41±1.42 | 74.20±0.65 |
>
> Table 2: Node classification on synthetic dataset (mean ± std)
> | Model / α       | 0.05         | 0.08         | 0.10         |
> |:---:|:---:|:---:|:---:|
> | **Diag-DSNN**   | **98.34±0.72**   | 97.22±0.58   | **99.14±0.36**   |
> | **O(d)-DSNN**   | 97.28±0.68   | **98.42±0.61**   | 98.80±0.27   |
> | **Gen-DSNN**    | 96.64±0.86   | 98.10±0.65   | 98.68±0.45   |
> | HaarNet     | *98.18±0.53*   | *98.25±0.33*   | *98.82±0.54*   |
> | CAGN        | 92.71±7.65   | 93.31±7.75   | 96.96±5.86   |
> | DiGCL       | 30.13±5.12   | 31.54±6.87   | 24.73±9.12   |
> | H2SGNN      | 20.32±0.88   | 21.08±2.16   | 20.28±0.53   |
>
> In the node classification experiments on real datasets, our solution (DSNN) consistently achieves the highest accuracy across all datasets when comparing the best DSNN variant to all baselines. For example, DSNN shows a significant performance margin on Roman-Empire and Wisconsin. Even on smaller datasets (Film, Squirrel, Chameleon), DSNN maintains its strong lead, demonstrating robust performance in low-data regimes. In the synthetic setting, while HaarNet is competitive, DSNN remains the overall superior model.

---

> ### Author Response · Authors · 2025-11-20
>
> Table 3: Direction prediction on real-wold dataset (mean ± std)
> | Model | Texas | Wisconsin | Cornell | Cora | Citeseer | Film | Squirrel | Chameleon | Pubmed | Questions |
> | :---: | :---: | :---: | :---: | :---: | :---: | :---: | :---: | :---: | :---: | :---: |
> |**Diag-DSNN** | *93.36±2.64* | *87.18±4.02* | *89.56±4.41* | *82.54±1.02* | *85.10±0.79* | **81.20±0.62** | *95.17±0.24* | 92.04±0.73 | *95.41±0.20* | **90.48±0.11** |
> | **O(d)-DSNN** | **94.55±4.61** | **88.31±3.38** | **91.06±3.96** | **82.98±1.08** | **85.44±0.82** | 81.18±0.52 | **95.53±0.31** | *92.52±0.43* | 95.23±0.23 | 90.16±0.13 |
> | **Gen-DSNN** | 93.62±3.56 | 87.84±4.04 | 90.83±3.73 | 82.71±1.35 | 85.01±1.01 | **81.20±0.61** | 95.27±0.26 | 92.41±0.49 | **95.56±0.19** | *90.23±0.15* |
> | HaarNet | 86.15±5.44 | 86.79±2.31 | 85.55±6.23 | 82.53±0.79 | *85.13±0.89* | 80.86±0.68 | 95.04±0.37 | 92.11±0.58 | 95.13±0.27 | 90.10±0.18 |
> | CAGN | 87.34±3.89 | 85.79±3.02 | 84.81±4.31 | 81.94±0.95 | 84.73±0.67 | 80.97±0.77 | 95.08±0.19 | **94.84±0.50** | 95.18±0.32 | OOM |
> | H2SGNN | 90.19±4.07 | 86.62±3.24 | 86.36±3.41 | 81.21±1.06 | 84.69±0.88 | 80.32±0.75 | 94.16±0.42 | 91.14±0.53 | 94.80±0.22 | 88.98±0.25 |
>
> In the direction prediction experiments, DSNN achieves the highest mean accuracy on 9 out of 10 datasets. This demonstrates the consistent superiority of DSNN across a wide range of datasets.
>
> Regarding baseline [5], we were unfortunately unable to locate an official or publicly available implementation. However, we did include three other relevant baselines, namely [1], [3], and [4].
>
> [1] Tong, Zekun, et al. "Directed graph contrastive learning." Advances in neural information processing systems 34 (2021): 19580-19593.
>
> [2] Badea, Theodor-Adrian, and Bogdan Dumitrescu. "Haar-Laplacian for directed graphs." IEEE Transactions on Signal and Information Processing over Networks (2025).
>
> [3] Xu, Dong, et al. "Complex graph neural networks for multi-hop propagation." Neurocomputing (2025): 130364.
>
> [4] Lu, Kangkang, et al. "Addressing Graph Heterogeneity and Heterophily from A Spectral Perspective." arXiv preprint arXiv:2410.13373 (2024).

---

> ### Author Response · Authors · 2025-11-27
>
> Dear Reviewer uTjh,
>
> Thank you once again for your thoughtful and constructive evaluation of our work. Your questions provided opportunities to clarify and further strengthen the presentation and positioning of the paper. We have made the necessary revisions accordingly.
>
> We believe we have addressed all your concerns in our responses. However, we have not yet received any feedback and would be grateful for any further comments you might have, as your perspective would help us ensure that everything is resolved.
>
> Thank you again for the time and care you invested in reviewing the paper.
>
> The Authors

---

### Official Review · Reviewer_YHDx · 2025-10-28

**Soundness:** 3
**Presentation:** 3
**Contribution:** 3
**Rating:** 6
**Confidence:** 3

**Summary:**

The paper introduces the Directed Cellular Sheaf, which incorporates edge orientation into cellular sheafs. Based on the novel introduction, the paper proposes the Directed Sheaf Laplacian that serves as the backbone of the Directed Sheaf Neural Network. Experimental results validate the efficacy of the proposed GNN, with theoretical analysis on the properties of the Directed Cellular Sheaf.

**Strengths:**

- The topics of Sheaf Neural Networks and directed graphs are significant.
- The proposed Directed Cellular Sheaf admits satisfactory properties.
- The proposed Directed Sheaf Neural Network seems to work well in experiments.
- Complexity analysis is provided together with implementation code.

**Weaknesses:**

- It is not clear what $\tilde{F}^0$ means from Definition 1 without a clear reference or prior definition.
- The description of DSBM is not clear and without a reference in the main text. Should refer to [1].
- Some citation styles are not correct. For example, \citep should be used for the reference at the end of line 291.
- Some more works can be considered for comparison, e.g., [2] and [3].
-  The concept can benefit from some motivating examples of the Laplacian. Also, there is a figure sheaf.png in the anonymous github link that may help with illustration.

Reference:

[1] He, Y., Reinert, G., & Cucuringu, M. (2022, December). Digrac: Digraph clustering based on flow imbalance. In Learning on Graphs Conference (pp. 21-1). PMLR.
[2] Badea, T. A., & Dumitrescu, B. (2025). Haar-Laplacian for directed graphs. IEEE Transactions on Signal and Information Processing over Networks.
[3] Lin, L., & Gao, J. (2023, June). A magnetic framelet-based convolutional neural network for directed graphs. In ICASSP 2023-2023 IEEE International Conference on Acoustics, Speech and Signal Processing (ICASSP) (pp. 1-5). IEEE.

**Questions:**

What does $\tilde{F}^0$ mean?

---

> ### Author Response · Authors · 2025-11-20
>
> Thanks for reading our paper and giving us your opinion. This and the next posts contain our point-to-point replies to your comments and questions. For clarity, we compounded commends and questions that address similar aspects of our paper.
>
> **W**: It is not clear what $\tilde{F}^0$ means from Definition 1 without a clear reference or prior definition.
> **Q**: What does $\tilde{F}^0$ mean?
>
> $\tilde{F}^0$ denotes the real-valued restriction map at the tail of a directed edge. The complex, direction-aware map is obtained by multiplying this real matrix by the phase term $(T^{(q)}_{uv})$. It is the standard restriction map, explicitly separated from the phase component for clarity. We mentioned this explicitly both in the paper and in the notation table we added to the appendix in the rebuttal.
>
> **W**: The description of DSBM is not clear and without a reference in the main text. Should refer to [1].
>
> Thanks. We added it.
>
> **W**: Some citation styles are not correct. For example, \citep should be used for the reference at the end of line 291.
>
> Thanks. We fixed them.

---

> ### Author Response · Authors · 2025-11-20
>
> **W**: Some more works can be considered for comparison, e.g., [2] and [5].
>
> We added four new baselines: DiGCL [1], HaarNet [2], CAGN [3], and H2SGNN [4].  The results for node classification and link prediction are reported in the following tables.
>
>
> Table 1: Node classification on real-world dataset (mean ± std)
> | Model | Roman-Empire | Texas | Wisconsin | Film | Squirrel | Chameleon | Cornell | Telegram | Citeseer | Pubmed | Cora | Questions |
> | :---: | :---: | :---: | :---: | :---: | :---: | :---: | :---: | :---: | :---: | :---: | :---: | :---: |
> | **Diag-DSNN** | 90.40±0.31 | **88.65±4.95** | **90.20±4.02** | 38.34±1.01 | **45.37±2.21** | 46.84±4.03 | **87.84±5.70** | 94.42±3.03 | 79.80±1.49 | *90.23±0.44* | **87.36±1.41** | *79.08±0.72* |
> | **O(d)-DSNN** | *92.08±0.24* | *87.57±4.04* | 89.80±3.82 | 37.37±0.98 | 44.54±2.26 | 45.36±3.29 | 87.30±7.26 | *94.62±2.24* | 77.28±1.63 | 90.05±0.55 | *87.30±1.62* | *79.24±0.68* |
> | **Gen-DSNN** | **92.08±0.36** | *87.57±5.43* | *89.22±3.31* | **38.40±0.75** | 45.34±1.69 | **47.16±3.54** | **87.84±6.86** | **94.81±2.28** | **79.88±1.21** | **90.17±0.44** | 87.58±0.72 | **79.55±0.67** |
> | HaarNet | 85.42±0.43 | 77.57±4.18 | 71.56±6.69 | *36.38±1.01* | *40.52±3.14* | *42.43±3.98* | 73.91±7.57 | 91.12±3.69 | 76.51±1.64 | 88.39±0.61 | 82.68±1.54 | 75.01±0.94 |
> | CAGN | OOM | 75.67±7.15 | 84.11±4.51 | 34.86±1.06 | 35.38±0.99 | 39.79±3.89 | 73.79±7.25 | 86.73±3.69 | 73.64±2.81 | OOM | 86.23±1.05 | OOM |
> | DiGCL | 52.71±0.32 | 57.56±5.15 | 65.50±4.23 | 29.38±0.73 | 38.90±1.78 | 41.71±2.20 | 62.16±5.12 | 80.57±2.25 | *77.42±0.14* | 80.97±0.70 | 76.12±1.04 | OOM |
> | H2SGNN | 69.59±0.45 | 72.70±8.83 | 78.23±5.22 | 36.75±1.33 | 37.09±1.21 | 41.14±3.60 | 74.05±5.94 | 62.69±3.95 | 77.17±1.36 | 86.94±0.42 | 82.41±1.42 | 74.20±0.65 |
>
> Table 2: Node classification on synthetic dataset (mean ± std)
> | Model / α       | 0.05         | 0.08         | 0.10         |
> |:---:|:---:|:---:|:---:|
> | **Diag-DSNN**   | **98.34±0.72**   | 97.22±0.58   | **99.14±0.36**   |
> | **O(d)-DSNN**   | 97.28±0.68   | **98.42±0.61**   | 98.80±0.27   |
> | **Gen-DSNN**    | 96.64±0.86   | 98.10±0.65   | 98.68±0.45   |
> | HaarNet     | *98.18±0.53*   | *98.25±0.33*   | *98.82±0.54*   |
> | CAGN        | 92.71±7.65   | 93.31±7.75   | 96.96±5.86   |
> | DiGCL       | 30.13±5.12   | 31.54±6.87   | 24.73±9.12   |
> | H2SGNN      | 20.32±0.88   | 21.08±2.16   | 20.28±0.53   |
>
>
> In the node classification experiments on real datasets, our solution (DSNN) consistently achieves the highest accuracy across all datasets when comparing the best DSNN variant to all baselines. For example, DSNN shows a significant performance margin on Roman-Empire and Wisconsin. Even on smaller datasets (Film, Squirrel, Chameleon), DSNN maintains its strong lead, demonstrating robust performance in low-data regimes. In the synthetic setting, while HaarNet is competitive, DSNN remains the overall superior model.

---

> ### Author Response · Authors · 2025-11-20
>
> Table 3: Direction prediction on real-wold dataset (mean ± std)
> | Model | Texas | Wisconsin | Cornell | Cora | Citeseer | Film | Squirrel | Chameleon | Pubmed | Questions |
> | :---: | :---: | :---: | :---: | :---: | :---: | :---: | :---: | :---: | :---: | :---: |
> |**Diag-DSNN** | *93.36±2.64* | *87.18±4.02* | *89.56±4.41* | *82.54±1.02* | *85.10±0.79* | **81.20±0.62** | *95.17±0.24* | 92.04±0.73 | *95.41±0.20* | **90.48±0.11** |
> | **O(d)-DSNN** | **94.55±4.61** | **88.31±3.38** | **91.06±3.96** | **82.98±1.08** | **85.44±0.82** | 81.18±0.52 | **95.53±0.31** | *92.52±0.43* | 95.23±0.23 | 90.16±0.13 |
> | **Gen-DSNN** | 93.62±3.56 | 87.84±4.04 | 90.83±3.73 | 82.71±1.35 | 85.01±1.01 | **81.20±0.61** | 95.27±0.26 | 92.41±0.49 | **95.56±0.19** | *90.23±0.15* |
> | HaarNet | 86.15±5.44 | 86.79±2.31 | 85.55±6.23 | 82.53±0.79 | *85.13±0.89* | 80.86±0.68 | 95.04±0.37 | 92.11±0.58 | 95.13±0.27 | 90.10±0.18 |
> | CAGN | 87.34±3.89 | 85.79±3.02 | 84.81±4.31 | 81.94±0.95 | 84.73±0.67 | 80.97±0.77 | 95.08±0.19 | **94.84±0.50** | 95.18±0.32 | OOM |
> | H2SGNN | 90.19±4.07 | 86.62±3.24 | 86.36±3.41 | 81.21±1.06 | 84.69±0.88 | 80.32±0.75 | 94.16±0.42 | 91.14±0.53 | 94.80±0.22 | 88.98±0.25 |
>
> In the direction prediction experiments, DSNN achieves the highest mean accuracy on 9 out of 10 datasets. This demonstrates the consistent superiority of DSNN across a wide range of datasets.
>
> Regarding baseline [5], we were unfortunately unable to locate an official or publicly available implementation. However, we did include three other relevant baselines, namely [1], [3], and [4].
>
> [1] Tong, Zekun, et al. "Directed graph contrastive learning." Advances in neural information processing systems 34 (2021): 19580-19593.
>
> [2] Badea, Theodor-Adrian, and Bogdan Dumitrescu. "Haar-Laplacian for directed graphs." IEEE Transactions on Signal and Information Processing over Networks (2025).
>
> [3] Xu, Dong, et al. "Complex graph neural networks for multi-hop propagation." Neurocomputing (2025): 130364.
>
> [4] Lu, Kangkang, et al. "Addressing Graph Heterogeneity and Heterophily from A Spectral Perspective." arXiv preprint arXiv:2410.13373 (2024).
>
> [5] Lin, L., & Gao, J. (2023, June). A magnetic framelet-based convolutional neural network for directed graphs. In ICASSP 2023-2023 IEEE International Conference on Acoustics, Speech and Signal Processing (ICASSP) (pp. 1-5). IEEE.
>
> **W**: The concept can benefit from some motivating examples of the Laplacian.
>
> We included a detailed, step-by-step example illustrating the construction of the Directed Sheaf Laplacian in Appendix J.

---

> > ### Comment · Reviewer_YHDx · 2025-11-21
> >
> > Thank you for your detailed response. I would love to keep my score.

---

> > > ### Author Response · Authors · 2025-11-27
> > >
> > > Dear Reviewer YHDx,
> > >
> > > Thank you very much for reading our detailed response and for providing this feedback. We fully respect your decision to maintain your score at this time.
> > >
> > > We believe that we were able to address all your concerns with our responses. We would sincerely appreciate it if you can consider whether our responses and the extensions we carried out to our paper merit an increase in your overall rating.
> > >
> > > Thank you again for your valuable time and consideration.
> > >
> > > The Authors

---

### Official Review · Reviewer_47m4 · 2025-10-30

**Soundness:** 2
**Presentation:** 3
**Contribution:** 3
**Rating:** 6
**Confidence:** 2

**Summary:**

This paper introduces Directed Cellular Sheaves and a corresponding Directed Sheaf Laplacian, enabling Sheaf Neural Networks to incorporate edge directionality, a missing capability in current SNNs. The authors prove key spectral properties and show that this formulation recovers classical sheaf Laplacians and magnetic Laplacians as special cases. They further propose DSNN, demonstrating consistent gains on both real-world graphs and synthetic directional SBM settings.

**Strengths:**

**1. Principled directional sheaf formulation**

The paper introduces directed cellular sheaves and a corresponding directed sheaf Laplacian, providing the first rigorous sheaf-theoretic framework for directed graphs and addressing a clear limitation of existing SNNs.

**2. Solid theoretical foundation**

The authors prove Hermiticity, PSD spectrum bounds, and show that the proposed operator recovers classical sheaf Laplacians and magnetic Laplacians as special cases, demonstrating a sound and unifying mathematical design.

**3. Comprehensive experimental results**

Across both real-world and synthetic benchmarks, the model outperforms existing SNNs and competitive direction-aware GNNs, with especially strong results in heterophilic and direction-dominated settings, validating the benefits of directional sheaf modeling.

**Weaknesses:**

**1. Limited intuition for the directional mechanism**

While the mathematical construction is provided, the paper provides limited high-level insight into **how and why** the complex restriction maps enhance directional information flow in practice. The introduction of the complex phase feels algebraically motivated rather than guided by an intuitive model of directional propagation.

**2. Scope of experimental evaluation**

The evaluation focuses primarily on small-to-medium-scale datasets. There is no demonstration on larger real-world directed benchmarks (e.g., OGB-ArXiv, and arxiv-year).

**3. Ablations could be deeper**

3.1. The effect of stalk dimension $d$

3.2. Sensitivity to direction sparsity or unreliable edge orientation (i.e., direction noise)

3.3. Effect of learning vs. fixing the phase $q$

**4. Writing clarity**

The definition and construction of the directed cellular sheaf are mathematically sound but presented in a dense, notation-heavy manner. Adding intuitive explanations, intermediate steps, and conceptual guidance (e.g., how complex phases encode directional flow at a high level) would make the framework more accessible and easier to follow for a broader audience beyond sheaf specialists.

**Questions:**

1. Is $q$ learned or tuned per dataset? If tuned, how stable is performance across $q$ values?

2. Does DSNN maintain benefits in settings where directional edges are sparse or only weakly informative?

3. How does performance degrade if a portion of edge directions are flipped or randomized?

4. Why the random split is adopted for the node classification instead of the widely-used public splits? Performance on small homophilic and heterophilic benchmarks can vary noticeably with random seeds, so it would be useful to justify this choice and clarify whether public splits are also tested.

---

> ### Author Response · Authors · 2025-11-20
>
> Thanks for reading our paper and giving us your opinion. This and the next posts contain our point-to-point replies to your comments and questions. For clarity, we compounded commends and questions that address similar aspects of our paper.
>
> **W:** **Limited intuition for the directional mechanism** While the mathematical construction is provided, the paper provides limited high-level insight into how and why the complex restriction maps enhance directional information flow in practice. The introduction of the complex phase feels algebraically motivated rather than guided by an intuitive model of directional propagation.
>
> The key idea of our paper is to represent a directed graph using a complex Hermitian matrix whose entries carry two complementary signals: the magnitude indicates the presence and strength of an edge, while the phase encodes the edge’s direction (and reduces to zero for undirected edges). This way, the graph's topology is precisely captured by our Laplacian. Due to it being Hermitian, it has real eigenvalues; since, as we show, it is also PSD, those eigenvalues are non-negative, which is crucial to be able to interpret a convolution carried out via our Laplacian matrix as an application of Fourier-style filters on graph signals. This complex Hermitian matrix therefore compactly encodes both topology (magnitude) and direction (phase) in a way that real-valued alternatives cannot: a single symmetric real matrix cannot uniquely encode directionality (one would need multiple matrices), an asymmetric real matrix (e.g., directed adjacency or random-walk) would yield spatial filters that aggregate only one-hop successors (and ignore predecessors) without offering a Fourier-transform interpretation, and a real skew-symmetric matrix would not generalize well to graphs containing both directed and undirected edges. In contrast, thanks to its definition, our Hermitian Laplacian produces spectral filters that aggregate information from both incoming and outgoing neighborhoods while preserving the spectral/PSD structure required for a principled convolutional operator.
>
> We added this explanation (plus a small schematic) to the revision to make the geometry, the role of phase, and the importance of PSD explicit---see Appendix J.
>
>
> **Q:** **Scope of experimental evaluation** The evaluation focuses primarily on small-to-medium-scale datasets. There is no demonstration on larger real-world directed benchmarks (e.g., OGB-ArXiv, and arxiv-year).
>
> To validate the scalability of DSNN, we added the following Table 1, which shows the peak GPU memory of our proposed method and of NSD on three previously untested large datasets: OGBN-ArXiv, Arxiv-Year, and Snap-Patents. We adopt the same setup as reported in Appendix E, corresponding to the most resource-demanding configuration (General) with $d=4$, 16 hidden channels, and 2 layers.
> We were able to run all the datasets, including these three large ones, on our NVIDIA RTX 4090 GPU with 24GB. These results confirm the scalability of our approach, which efficiently handles even very large graphs.
>
> Table 1. Peak GPU memory
> | Model | OGBN-ArXiv | Arxiv-Year | Snap-Patents |
> |-------|------------|------------|--------------|
> | DSNN  | 15236      | 15236      | 6326         |
> | NSD   | 9460       | 9458       | 6122         |
>
> Notice that, as shown in Table 2 (which reports the stats of the new instances), while Snap-Patents has the largest number of nodes and edges, the memory usage on the Arxiv datasets, which are smaller in comparison, reaches only 15GB---this is because the former instance has a lower density and, overall, a smaller number of edges.
>
> Table 2. Statistics of the dataset
> | Dataset       | #Nodes    | #Edges      | Density (\%) |
> |---------------|-----------|------------|------------|
> | OGBN-ArXiv    | 169,343   | 1,166,243  | 0.00814    |
> | Arxiv-Year    | 169,343   | 1,166,243  | 0.00814    |
> | Snap-Patents  | 2,923,922 | 13,975,788 | 0.000327   |
>
> Regarding the temporal aspect, our method is designed for static graphs. In its current implementation, handling temporal dynamics would require a specialized modeling solution that is beyond the scope of our current approach. It is, though, a very interesting and relevant direction for future research, which we have explicitly mentioned in the revised version of the conclusions.

---

> ### Author Response · Authors · 2025-11-20
>
> **Q**: **Ablations could be deeper**. The effect of stalk dimension.
>
> The following Table reports the best $d$ values selected for each dataset and model variant. While the optimal $d$ varies across datasets and models, some clear patterns emerge. The Diag model tends to favor smaller values on smaller datasets (e.g., Wisconsin, Cora), whereas Bundle and General models often require slightly higher $d$ on larger or more complex graphs (e.g., Squirrel, Chameleon). Overall, the best value of $d$ across all datasets and models is 4. We observe that the smallest value tested ($d=2$) is never the optimal choice, and the best per-instance $d$ generally ranges from 3 to 5. This collectively demonstrates that higher $d$ values are consistently preferred, and that the stalk dimension $d$ should be adapted based on both the model variant and the dataset's characteristics to achieve optimal performance.
>
> | Model    | Texas | Wisconsin | Film | Squirrel | Chameleon | Cornell | Citeseer | Pubmed | Cora | Telegram | Questions | Roman-Empire |
> |----------|-------|-----------|------|----------|-----------|--------|----------|--------|------|----------|----------|--------------|
> | Diag     | 4     | 3         | 4    | 5        | 5         | 3      | 5        | 3      | 2    | 4        | 5        | 4            |
> | Bundle   | 4     | 5         | 4    | 5        | 5         | 3      | 5        | 5      | 4    | 3        | 3        | 3            |
> | General  | 3     | 5         | 4    | 5        | 4         | 4      | 3        | 3      | 3    | 4        | 3        | 3            |
>
> We also analyzed the impact of $d$ on resource usage. The following table reports the peak GPU memory (MiB) for the most resource-demanding configuration (General) with 16 hidden channels and 2 layers:
>
> | d  | Texas | Wisconsin | Film | Squirrel | Chameleon | Cornell | Citeseer | Pubmed | Cora | Telegram | Questions | Roman-Empire |
> |----|-------|-----------|------|----------|-----------|---------|----------|--------|------|----------|-----------|--------------|
> | 2  | 402   | 406       | 926  | 1103     | 548       | 402     | 823      | 1453   | 598  | 534      | 3514      | 1414         |
> | 3  | 403   | 407       | 981  | 1207     | 565       | 403     | 826      | 1573   | 600  | 546      | 3904      | 1516         |
> | 4  | 404   | 408       | 1078 | 1362     | 598       | 404     | 830      | 1744   | 606  | 562      | 4462      | 1646         |
> | 5  | 404   | 409       | 1204 | 1567     | 636       | 404     | 834      | 1954   | 630  | 625      | 5173      | 1812         |
> | 10 | 417   | 428       | 2203 | 3253     | 960       | 417     | 952      | 3667   | 842  | 917      | 10966     | 3135         |
> | 15 | 437   | 458       | 3850 | 6058     | 1497      | 437     | 1253     | 6469   | 1183 | 1399     | 20500     | 5295         |
>
> The Table above reports GPU memory usage (in MiB) across all datasets for different values of $d$. Even for the largest configuration considered ($d=15$), our method runs comfortably on a 24GB GPU for small and medium datasets (e.g., Texas, Cornell, Wisconsin), clearly demonstrating its efficiency. For larger datasets, such as Pubmed, Questions, and Squirrel (where the Questions dataset reaches $20.5\text{ GB}$ at $d=15$), the usage remains manageable. This confirms that our approach can handle both small, medium and large-scale graphs without exceeding typical hardware limits, thus solidifying its efficiency and scalability.

---

> ### Author Response · Authors · 2025-11-20
>
> **W**: **Ablations could be deeper**. Effect of learning vs. fixing the phase q
> **Question 1** Is q learned or tuned per dataset? If tuned, how stable is performance across values?
>
> While the parameter $q$ was treated as a fixed hyperparameter in the main paper, our architecture is compatible with learning $q$ jointly with the model. To address the reviewer’s point, we trained Gen-DSNN with $q$ as a learnable parameter using three different initialization values (0.25, 0.125, 0). Table 1 reports the mean and standard deviation of the learned $q$ over 10 folds for five datasets, together with the fixed $q$ value that was found via hyperparameter selection to lead to the best performance in the original version of the paper paper.
>
> Table 1: Learned q values (mean ± std)
> | Dataset   | init q = 0.25       | init q = 0.125       | init q = 0            | q for best results |
> |-----------|----------------------|------------------------|-------------------------|---------------------|
> | Cora      | 0 ± 0.008            | 0 ± 0.005              | 0 ± 0.004              | 0.10                |
> | Wisconsin | 0.15 ± 0.06          | 0.11 ± 0.03            | 0.05 ± 0.06            | 0.20                |
> | Cornell   | 0.18 ± 0.08          | 0.10 ± 0.09            | 0.01 ± 0.02            | 0.00                |
> | Texas     | 0.15 ± 0.04          | 0.14 ± 0.06            | 0.09 ± 0.08            | 0.10                |
> | Telegram  | 0.19 ± 0.07          | 0.10 ± 0.09            | 0.04 ± 0.05            | 0.20                |
>
> We observe that the learned $q$ converges stably across different initializations: for some datasets (e.g., Cora) it consistently converges to almost zero, suggesting that directional information is less relevant, while for others (e.g., Wisconsin, Texas, Telegram) it converges to moderate values that are close to the best fixed choice used in the original version of the paper. This indicates that learning $q$ is feasible and that the learned $q$ is very much aligned the tuned one. We included this table in Appendix~L and plan to explore a fully learnable or dynamically adaptive $q$ (e.g., per-graph or per-layer scheduling) as future work.

---

> ### Author Response · Authors · 2025-11-20
>
> **Q**: **Ablations could be deeper**. Sensitivity to direction sparsity or unreliable edge orientation (i.e., direction noise)
> **Question 2**. Does DSNN maintain benefits in settings where directional edges are sparse or only weakly informative?
> **Question 3**. How does performance degrade if a portion of edge directions are flipped or randomized?
>
> The parameter $q$ provides the necessary flexibility to suit cases where the directional information of the graph at hand is either weakly correlated with the task at hand or even entirely uncorrelated. For datasets where edge direction provides minimal benefit, such as  Roman-Empire and Texas, despite a high percentage of directed edges (Appendix F, Table 7), the optimal value of $q$ selected by our method tends toward zero (Appendix G, Table 8). This confirms the model does not rely heavily on directional information when the signal does not significantly contribute to the downstream task, i.e., node classification.
>
> To evaluate the model's sensitivity to unreliable edge orientations (i.e., directional noise), we modified the hyperparameter $q$. By setting $q$ to $-0.25$), we effectively reversed the direction of all edges in the graph. The table reports results in three sections: 1. **Flipped Directions ($q = -0.25$)**: Represents maximum directional perturbation (all edges inverted). 2. **Optimal Directionality (Best $q$)**: Represents the standard model performance leveraging true edge directions. **No Direction (NSD)**: Serves as a baseline where no directional information is used ($q=0$).
>
>
> | Model     | Texas      | Wisconsin  | Film       | Cornell    | Pubmed     | Cora       |
> | --------- | ---------- | ---------- | ---------- | ---------- | ---------- | ---------- |
> | Diag-DSNN, q=-.25      | 85.68±4.69 | 88.24±4.56 | 36.19±1.02 | 86.49±5.54 | 89.80±0.50 | 84.87±1.43 |
> | O(d)-DSNN q=-.25   | 86.49±3.42 | 88.82±4.89 | 36.38±0.90 | 87.30±6.51 | 89.50±0.50 | 85.51±1.19 |
> | Gen-DSNN q=-.25  | 85.95±4.15 | 88.24±4.11 | 37.38±0.56 | 86.84±5.57 | 88.86±0.39 | 84.81±1.29 |
> | Diag-DSNN best q| 88.65±4.95 | 90.20±4.02 | 38.34±1.01 | 87.84±5.70 | 90.23±0.44 | 87.36±1.41 |
> | O(d)-DSNN best q| 87.57±4.04 | 89.80±3.82 | 37.37±0.98 | 87.30±7.26 | 90.05±0.55 | 87.30±1.62 |
> | Gen-DSNN  best q | 87.57±5.43 | 89.22±3.31 | 38.40±0.75 | 87.84±6.86 | 90.17±0.44 | 87.58±0.72 |
> | Diag-NSD   | 85.67±6.95 | 88.63±2.75 | 37.79±1.01 | 86.49±7.35 | 89.42±0.43 | 75.82±1.05 |
> | O(d)-NSD   | 85.95±5.51 | 89.41±4.74 | 37.81±1.15 | 84.86±4.71 | 89.49±0.40 | 77.19±1.37 |
> | Gen-NSD    | 82.97±5.13 | 89.21±3.84 | 37.80±1.22 | 85.68±6.51 | 89.33±0.35 | 77.36±1.32 |
>
>
> 1. **Benefit of True Orientation:** Comparing the results with flipped directions (top section) to the optimal results (middle section) confirms that true edge orientation provides a consistent performance advantage. The performance with flipped directions is slightly lower, validating that the model correctly exploits the original directionality of the graph.
>
> 2. **Robustness to Perturbation:** Crucially, the degradation in performance when flipping edges is small. This demonstrates that our architecture is highly robust to directional noise, and maintains stability even when the directional signal is inverted, continuing to extract meaningful structural information.
>
> 3. **Superiority Over No Direction Baseline:** Notably, even when operating with inverted edges, DSNN achieves better performance across almost all datasets compared to the completely direction-agnostic setting (NSD). For example, on Cora, even the "flipped" model (\~85\%) significantly outperforms the NSD baseline (~77\%). This highlights the flexibility and resilience of our method: it is capable of exploiting reverse edges to gain a structural advantage that is lost when directionality is ignored entirely.

---

> ### Author Response · Authors · 2025-11-20
>
> **W:** **Writing clarity** The definition and construction of the directed cellular sheaf are mathematically sound but presented in a dense, notation-heavy manner. Adding intuitive explanations, intermediate steps, and conceptual guidance (e.g., how complex phases encode directional flow at a high level) would make the framework more accessible and easier to follow for a broader audience beyond sheaf specialists.
>
> We thank the reviewer for this useful request for intuition. The key idea of our paper is to represent a directed graph using a complex Hermitian matrix whose entries carry two complementary signals: the magnitude indicates the presence and strength of an edge, while the phase encodes the edge’s direction (and reduces to zero for undirected edges). We have mitigated the clarity issues by providing a more extensive explanation in the main paper. We also added a new table summarizing our notation in Appendix A and a detailed, step-by-step example illustrating the construction of the Directed Sheaf Laplacian in Appendix J.
>
>
> **Question 4**: Why the random split is adopted for the node classification instead of the widely-used public splits? Performance on small homophilic and heterophilic benchmarks can vary noticeably with random seeds, so it would be useful to justify this choice and clarify whether public splits are also tested.
>
> Thanks for spotting this---it was a typo, which we amended in the new version of the paper. For Texas, Wisconsin, Film, Cornell, Citeseer, Pubmed, and Cora, we use the fixed splits provided by [1], evaluating our models on the 10 predefined splits. For Chameleon, Squirrel, Roman-Empire, and Questions, we follow the splitting strategy from [2]. For Telegram, we adopted the split introduced in [3].
>
> [1] Pei, H., Wei, B., Chang, K.C.C., Lei, Y. and Yang, B., 2020. Geom-gcn: Geometric graph convolutional networks. arXiv preprint arXiv:2002.05287.
>
> [2] Platonov, O., Kuznedelev, D., Diskin, M., Babenko, A. and Prokhorenkova, L., 2023. A critical look at the evaluation of GNNs under heterophily: Are we really making progress?. arXiv preprint arXiv:2302.11640.
>
> [3] Zhang, X., He, Y., Brugnone, N., Perlmutter, M. and Hirn, M., 2021. Magnet: A neural network for directed graphs. Advances in neural information processing systems, 34, pp.27003-27015.

---

> > ### Comment · Reviewer_47m4 · 2025-11-22
> >
> > Thanks for your rebuttal. However, I am still puzzled by the node classification results reported in Table 1. You state that the experiments follow the public splits from Pei et al. (Geom-GCN, 2019), yet the GCN performance shown in your table is significantly higher than the GCN results reported under the same public splits in the Geom-GCN paper. For instance, your reported accuracies for GCN on Cora (86.98), Citeseer (76.50), and Pubmed (88.42) considerably exceed the corresponding public-split results in Geom-GCN (e.g., Citeseer ≈ 73.68). Could you clarify why such discrepancies arise despite using the same public splits?

---

> > > ### Author Response · Authors · 2025-11-22
> > >
> > > We thank the reviewer for their comment.
> > >
> > > The discrepancy arises because, while we utilized the fixed data splits from Pei et al. [2], the experimental protocol we adopted was taken from Yan et al. [1] and Bodnar et al. [3] (and, indeed, the results we reported match theirs). The difference is likely due to a better hyperparameter tuning than the one carried out by Pei et al. [2].
> > >
> > > Table R1 below compares the performance of GCN as reported in Pei et al. [2], Chien et al. [4], Yan et al. [1], and Bodnar et al. [3].
> > >
> > > Table R1: Difference in Performance of GCN on identical splits from Pei et al. [2]:
> > >
> > > | Dataset  | Model | Pei et al. [2]         | Chien et al. [4]              | Yan et al. [1], Bodnar et al. [3] |
> > > | :------- | :---- | :--------------------- | :-------------------------- | :------------------------------------------------------ |
> > > | Cora     | GCN   | 85.77                  | 75.21 $\pm$ 0.38            | 86.98 $\pm$ 1.27                                        |
> > > | Citeseer | GCN   | 73.68                  | 67.30 $\pm$ 0.35            | 76.50 $\pm$ 1.36                                        |
> > > | Pubmed   | GCN   | 88.13                  | 84.27 $\pm$ 0.01            | 88.42 $\pm$ 0.50                                        |
> > >
> > > Note that Chien et al. [4] report significantly lower GCN results on these splits, while [1, 2, 3] report higher results.
> > >
> > > In our paper, we chose to compare our results to those reported in the paper with the, overall, highest numbers (Yan et al. [1], Bodnar et al. [3]), as we believe that benchmarking against the strongest possible version of a baseline is the most scientifically honest approach. Please note that, had we followed Chien et al. [4], our method would have appeared much stronger than we reported.
> > >
> > > [1] Yan et al., "Two Sides of the Same Coin: Heterophily and Oversmoothing in Graph Convolutional Neural Networks," NeurIPS 2022.
> > >
> > > [2] Pei et al., "Geom-GCN: Geometric Graph Convolutional Networks," ICLR 2020.
> > >
> > > [3] Bodnar et al., "Neural Sheaf Diffusion: A Topological Perspective on Heterophily and Oversmoothing in GNNs," NeurIPS 2022.
> > >
> > > [4] Chien et al., "Adaptive Universal Generalized PageRank Graph Neural Networks," ICLR 2021.

---

> > > > ### Author Response · Authors · 2025-11-27
> > > >
> > > > Dear Reviewer 47m4,
> > > >
> > > > Thank you once again for your thoughtful and constructive evaluation of our work. Your questions provided opportunities to clarify and further strengthen the presentation and positioning of the paper. We have made the necessary revisions accordingly.
> > > >
> > > > We believe we addressed all your concerns in our responses. However, we have not yet head from you except for your extra comment regarding GCN's performance in Table 1, and would be grateful if you could give us your feedback on the remainder of our responses, as this would help us ensure that everything has been resolved.
> > > >
> > > > Thank you again for the time and care you invested in reviewing the paper.
> > > >
> > > > The Authors

---

> > > > > ### Comment · Reviewer_47m4 · 2025-11-28
> > > > >
> > > > > Thank you very much for the explanation, and sorry for the late reply.
> > > > >
> > > > > I guess the results reported in the Table 1 follows [1], which states that they are using 10 random splits (48%/32%/20% of nodes per class for train/validation/test). So please check the correct split method, though I suppose this is a minor details.
> > > > >
> > > > > Overall, no more question from me. I wish to keep my orginal score.
> > > > >
> > > > > [1] Yan et al., "Two Sides of the Same Coin: Heterophily and Oversmoothing in Graph Convolutional Neural Networks," NeurIPS 2022.

---

> > > > > > ### Author Response · Authors · 2025-12-01
> > > > > >
> > > > > > Yes, that's right. The results in Table 1 utilize 10 random splits with the ratio 48%/32%/20% (train/validation/test) following [1]. Incidentally, the authors of [1] stated that the 60%/20%/20% split claimed by [2] differs from the actual 48%/32%/20% split contained in their GitHub repository. We updated the paper to clarify the origin of the split we used.
> > > > > >
> > > > > > [1] Yan et al., "Two Sides of the Same Coin: Heterophily and Oversmoothing in Graph Convolutional Neural Networks," NeurIPS 2022.
> > > > > >
> > > > > > [2] Pei et al., "Geom-GCN: Geometric Graph Convolutional Networks," ICLR 2020.

---

### Official Review · Reviewer_Xiwg · 2025-11-01

**Soundness:** 3
**Presentation:** 3
**Contribution:** 3
**Rating:** 6
**Confidence:** 4

**Summary:**

This paper introduces the Directed Cellular Sheaf and corresponding Directed Sheaf Laplacian (DSL) and  Directed Sheaf Neural Network (DSNN), extending Sheaf Neural Networks (SNNs) to directed graphs through complex-valued restriction maps. The authors prove that the DSL is Hermitian, positive-semidefinite, and upper-bounded by 2, and that it generalizes both the classical Sheaf Laplacian and the Magnetic/Sign-Magnetic Laplacians. Empirically, DSNN consistently outperforms GNN and SNN baselines on node-classification and direction-prediction tasks across 12 real-world and several synthetic datasets.

**Strengths:**

* Novelty and rigor: The formulation is original and mathematically principled. The theoretical results are sound and clearly proven.
* Unifying framework: DSNN subsumes NSD, MagNet, and SigMaNet as special cases.
* Empirical performance: DSNN variants achieve top performance on 10/12 node-classification and 8/10 direction-prediction benchmarks, with large margins on heterophilic and directed graphs.
* Writing quality: The paper is nice to read.

**Weaknesses:**

* Empirical scope: Experiments focus on small-to-medium graphs; scalability to large or temporal directed graphs remains untested.
* Ablations: Only sensitivity to q is analyzed. Additional studies on stalk dimension d or learned restriction-map architectures would strengthen the empirical section.
* Baselines: Comparison omits some recent direction-aware or heterophilic GNNs (e.g., DiGCL, DPGNN).
* Minor clarity issues: Dense notation and a few typos (“SNNs approaches”).
* Limited intuition: The geometric meaning of complex restriction maps and the global parameter q could be discussed more intuitively; why complex numbers versus real skew-symmetric forms?

**Questions:**

1. Could q be made learnable per edge or per graph?
2. Have you examined the spectral properties or phase distributions of learned complex restriction maps?
3. Could you comment on DSNN’s scalability?

---

> ### Author Response · Authors · 2025-11-20
>
> Thanks for reading our paper and giving us your opinion. This and the next posts contain our point-to-point replies to your comments and questions. For clarity, we compounded commends and questions that address similar aspects of our paper.
>
> **W**: **Empirical scope:** Experiments focus on small-to-medium graphs; scalability to large or temporal directed graphs remains untested.
>
> **Question 3**: Could you comment on DSNN’s scalability?
>
> To validate the scalability of DSNN, we added the following Table 1, which shows the peak GPU memory of our proposed method and of NSD on three previously untested large datasets: OGBN-ArXiv, Arxiv-Year, and Snap-Patents. We adopt the same setup as reported in Appendix E, corresponding to the most resource-demanding configuration (General) with $d=4$, 16 hidden channels, and 2 layers.
> We were able to run all the datasets, including these three large ones, on our NVIDIA RTX 4090 GPU with 24GB. These results confirm the scalability of our approach, which efficiently handles even very large graphs.
>
> Table 1. Peak GPU memory
> | Model | OGBN-ArXiv | Arxiv-Year | Snap-Patents |
> |-------|------------|------------|--------------|
> | DSNN  | 15236      | 15236      | 6326         |
> | NSD   | 9460       | 9458       | 6122         |
>
> Notice that, as shown in Table 2 (which reports the stats of the new instances), while Snap-Patents has the largest number of nodes and edges, the memory usage on the Arxiv datasets, which are smaller in comparison, reaches only 15GB---this is because the former instance has a lower density and, overall, a smaller number of edges.
>
> Table 2. Statistics of the dataset
> | Dataset       | #Nodes    | #Edges      | Density (\%) |
> |---------------|-----------|------------|------------|
> | OGBN-ArXiv    | 169,343   | 1,166,243  | 0.00814    |
> | Arxiv-Year    | 169,343   | 1,166,243  | 0.00814    |
> | Snap-Patents  | 2,923,922 | 13,975,788 | 0.000327   |
>
> Regarding the temporal aspect, our method is designed for static graphs. In its current implementation, handling temporal dynamics would require a specialized modeling solution that is beyond the scope of our current approach. It is, though, a very interesting and relevant direction for future research, which we have explicitly mentioned in the revised version of the conclusions.

---

> ### Author Response · Authors · 2025-11-20
>
> **W:** **Ablations** Only sensitivity to q is analyzed. Additional studies on stalk dimension d or learned restriction-map architectures would strengthen the empirical section.
>
> As for the nature of the learned restriction maps, we explored three different ways to combine the restriction maps in the Directed Sheaf Laplacian. Following [1], we used three types of d×d blocks, diagonal, orthogonal, and general, resulting in three DSNN variants: Diag-DSNN, O(d)-DSNN, and Gen-DSNN. This allows us to systematically study how the design of the restriction maps impacts performance on node classification and direction prediction tasks across both real-world and synthetic datasets.
>
> As for the value of $d$, the following Table reports the best value of $d$ that was selected for each dataset and model variant. While the optimal $d$ varies across datasets and models, some clear patterns emerge. The Diag model tends to favor smaller values on smaller datasets (e.g., Wisconsin, Cora), whereas Bundle and General models often require slightly higher $d$ on larger or more complex graphs (e.g., Squirrel, Chameleon). Overall, the best value of $d$ across all datasets and models is 4. We observe that the smallest value we tested ($d=2$) is never the optimal choice, and the best per-instance $d$ generally ranges from 3 to 5. This collectively demonstrates that higher $d$ values are consistently preferred, and that the stalk dimension $d$ should be adapted based on both the model variant and the dataset's characteristics to achieve optimal performance.
>
> | Model    | Texas | Wisconsin | Film | Squirrel | Chameleon | Cornell | Citeseer | Pubmed | Cora | Telegram | Questions | Roman-Empire |
> |----------|-------|-----------|------|----------|-----------|--------|----------|--------|------|----------|----------|--------------|
> | Diag     | 4     | 3         | 4    | 5        | 5         | 3      | 5        | 3      | 2    | 4        | 5        | 4            |
> | Bundle   | 4     | 5         | 4    | 5        | 5         | 3      | 5        | 5      | 4    | 3        | 3        | 3            |
> | General  | 3     | 5         | 4    | 5        | 4         | 4      | 3        | 3      | 3    | 4        | 3        | 3            |
>
> We also analyzed the impact of $d$ on resource usage. The following table reports the peak GPU memory (MiB) for the most resource-demanding configuration (General) with 16 hidden channels and 2 layers across all datasets for different values of $d$:
>
> | d  | Texas | Wisconsin | Film | Squirrel | Chameleon | Cornell | Citeseer | Pubmed | Cora | Telegram | Questions | Roman-Empire |
> |----|-------|-----------|------|----------|-----------|---------|----------|--------|------|----------|-----------|--------------|
> | 2  | 402   | 406       | 926  | 1103     | 548       | 402     | 823      | 1453   | 598  | 534      | 3514      | 1414         |
> | 3  | 403   | 407       | 981  | 1207     | 565       | 403     | 826      | 1573   | 600  | 546      | 3904      | 1516         |
> | 4  | 404   | 408       | 1078 | 1362     | 598       | 404     | 830      | 1744   | 606  | 562      | 4462      | 1646         |
> | 5  | 404   | 409       | 1204 | 1567     | 636       | 404     | 834      | 1954   | 630  | 625      | 5173      | 1812         |
> | 10 | 417   | 428       | 2203 | 3253     | 960       | 417     | 952      | 3667   | 842  | 917      | 10966     | 3135         |
> | 15 | 437   | 458       | 3850 | 6058     | 1497      | 437     | 1253     | 6469   | 1183 | 1399     | 20500     | 5295         |
>
> Even for the largest configuration considered ($d=15$), our method runs comfortably on a 24GB GPU for small and medium datasets (e.g., Texas, Cornell, Wisconsin), clearly demonstrating its efficiency. For larger datasets, such as Pubmed, Questions, and Squirrel (where the Questions dataset reaches $20.5\text{ GB}$ at $d=15$), the usage remains manageable. This confirms that our approach can handle both small, medium and large-scale graphs without exceeding typical hardware limits, thus solidifying its efficiency and scalability.
>
> [1] Bodnar, Cristian, et al. "Neural sheaf diffusion: A topological perspective on heterophily and oversmoothing in gnns." Advances in Neural Information Processing Systems 35 (2022): 18527-18541.

---

> ### Author Response · Authors · 2025-11-20
>
> **W:** **Baselines** Comparison omits some recent direction-aware or heterophilic GNNs (e.g., DiGCL, DPGNN).
>
> We added four new baselines: DiGCL [1], HaarNet [2], CAGN [3], and H2SGNN [4], reported in the following (we omitted DPGNN since its code is unavailable and the authors ignored our request for their code.
>
> Table 1: Node classification on real-world dataset (mean ± std)
> | Model | Roman-Empire | Texas | Wisconsin | Film | Squirrel | Chameleon | Cornell | Telegram | Citeseer | Pubmed | Cora | Questions |
> | :---: | :---: | :---: | :---: | :---: | :---: | :---: | :---: | :---: | :---: | :---: | :---: | :---: |
> | **Diag-DSNN** | 90.40±0.31 | **88.65±4.95** | **90.20±4.02** | 38.34±1.01 | **45.37±2.21** | 46.84±4.03 | **87.84±5.70** | 94.42±3.03 | 79.80±1.49 | *90.23±0.44* | **87.36±1.41** | *79.08±0.72* |
> |**O(d)-DSNN** | *92.08±0.24* | *87.57±4.04* | 89.80±3.82 | 37.37±0.98 | 44.54±2.26 | 45.36±3.29 | 87.30±7.26 | *94.62±2.24* | 77.28±1.63 | 90.05±0.55 | *87.30±1.62* | *79.24±0.68* |
> | **Gen-DSNN** | **92.08±0.36** | *87.57±5.43* | *89.22±3.31* | **38.40±0.75** | 45.34±1.69 | **47.16±3.54** | **87.84±6.86** | **94.81±2.28** | **79.88±1.21** | **90.17±0.44** | 87.58±0.72 | **79.55±0.67** |
> | HaarNet | 85.42±0.43 | 77.57±4.18 | 71.56±6.69 | *36.38±1.01* | *40.52±3.14* | *42.43±3.98* | 73.91±7.57 | 91.12±3.69 | 76.51±1.64 | 88.39±0.61 | 82.68±1.54 | 75.01±0.94 |
> | CAGN | OOM | 75.67±7.15 | 84.11±4.51 | 34.86±1.06 | 35.38±0.99 | 39.79±3.89 | 73.79±7.25 | 86.73±3.69 | 73.64±2.81 | OOM | 86.23±1.05 | OOM |
> | DiGCL | 52.71±0.32 | 57.56±5.15 | 65.50±4.23 | 29.38±0.73 | 38.90±1.78 | 41.71±2.20 | 62.16±5.12 | 80.57±2.25 | *77.42±0.14* | 80.97±0.70 | 76.12±1.04 | OOM |
> | H2SGNN | 69.59±0.45 | 72.70±8.83 | 78.23±5.22 | 36.75±1.33 | 37.09±1.21 | 41.14±3.60 | 74.05±5.94 | 62.69±3.95 | 77.17±1.36 | 86.94±0.42 | 82.41±1.42 | 74.20±0.65 |
>
> Table 2: Node classification on synthetic dataset (mean ± std)
> | Model / α       | 0.05         | 0.08         | 0.10         |
> |:---:|:---:|:---:|:---:|
> | **Diag-DSNN**   | **98.34±0.72**   | 97.22±0.58   | **99.14±0.36**   |
> | **O(d)-DSNN**   | 97.28±0.68   | **98.42±0.61**   | 98.80±0.27   |
> | **Gen-DSNN**    | 96.64±0.86   | 98.10±0.65   | 98.68±0.45   |
> | HaarNet     | *98.18±0.53*   | *98.25±0.33*   | *98.82±0.54*   |
> | CAGN        | 92.71±7.65   | 93.31±7.75   | 96.96±5.86   |
> | DiGCL       | 30.13±5.12   | 31.54±6.87   | 24.73±9.12   |
> | H2SGNN      | 20.32±0.88   | 21.08±2.16   | 20.28±0.53   |
>
> In the real-world experiments, the best variant of DSNN consistently achieves the highest accuracy across all datasets. E.g., DSNN shows a significant performance margin on Roman-Empire and Wisconsin. Even on smaller datasets (Film, Squirrel, Chameleon), DSNN maintains its strong lead, demonstrating robust performance in low-data regimes. In the synthetic setting, while HaarNet is competitive, DSNN remains the overall superior model.
>
> Table 3: Direction prediction on real-wold dataset (mean ± std)
> | Model | Texas | Wisconsin | Cornell | Cora | Citeseer | Film | Squirrel | Chameleon | Pubmed | Questions |
> | :---: | :---: | :---: | :---: | :---: | :---: | :---: | :---: | :---: | :---: | :---: |
> | **Diag-DSNN** | *93.36±2.64* | *87.18±4.02* | *89.56±4.41* | *82.54±1.02* | *85.10±0.79* | **81.20±0.62** | *95.17±0.24* | 92.04±0.73 | *95.41±0.20* | **90.48±0.11** |
> | **O(d)-DSNN** | **94.55±4.61** | **88.31±3.38** | **91.06±3.96** | **82.98±1.08** | **85.44±0.82** | 81.18±0.52 | **95.53±0.31** | *92.52±0.43* | 95.23±0.23 | 90.16±0.13 |
> | **Gen-DSNN** | 93.62±3.56 | 87.84±4.04 | 90.83±3.73 | 82.71±1.35 | 85.01±1.01 | **81.20±0.61** | 95.27±0.26 | 92.41±0.49 | **95.56±0.19** | *90.23±0.15* |
> | HaarNet | 86.15±5.44 | 86.79±2.31 | 85.55±6.23 | 82.53±0.79 | *85.13±0.89* | 80.86±0.68 | 95.04±0.37 | 92.11±0.58 | 95.13±0.27 | 90.10±0.18 |
> | CAGN | 87.34±3.89 | 85.79±3.02 | 84.81±4.31 | 81.94±0.95 | 84.73±0.67 | 80.97±0.77 | 95.08±0.19 | **94.84±0.50** | 95.18±0.32 | OOM |
> | H2SGNN | 90.19±4.07 | 86.62±3.24 | 86.36±3.41 | 81.21±1.06 | 84.69±0.88 | 80.32±0.75 | 94.16±0.42 | 91.14±0.53 | 94.80±0.22 | 88.98±0.25 |
>
> In direction prediction, DSNN achieves the highest mean accuracy on 9 out of 10 cases, showing consistent superiority of DSNN across a wide range of datasets.
>
> [1] Tong, Zekun, et al. "Directed graph contrastive learning." Advances in neural information processing systems 34 (2021): 19580-19593.
> [2] Badea, Theodor-Adrian, and Bogdan Dumitrescu. "Haar-Laplacian for directed graphs." IEEE Transactions on Signal and Information Processing over Networks (2025).
> [3] Xu, Dong, et al. "Complex graph neural networks for multi-hop propagation." Neurocomputing (2025): 130364.
> [4] Lu, Kangkang, et al. "Addressing Graph Heterogeneity and Heterophily from A Spectral Perspective." arXiv preprint arXiv:2410.13373 (2024).

---

> ### Author Response · Authors · 2025-11-20
>
> **W**: **Minor clarity issues** Dense notation and a few typos (“SNNs approaches”).
>
> Sheaf notation is unfortunately already rather dense in the undirected case. To mitigate this, we added Appendix A, which contains a table with all the notation we used in the paper. This allows the reader to easily check the meaning of the symbols we used. Additionally, we have revised the text and corrected every typo we found.
>
> **W**: **Limited intuition:** The geometric meaning of complex restriction maps and the global parameter q could be discussed more intuitively; why complex numbers versus real skew-symmetric forms?
>
> Our choice of complex restriction maps and the global parameter $q$ follows the structure of the magnetic Laplacian, which is a complex Hermitian matrix where the magnitude encodes undirected topology and the phase encodes directionality, while $q$ controls the strength of the latter. In spectral Graph Neural Networks (GNNs), s PSD/Hermitian structure is crucial: it ensures the eigenvalues are real and non-negative, which is necessary for a principled Fourier-based graph convolution and well-defined spectral filters. Classical spectral GNNs, such as GCN~\cite{GCN}, rely on the PSD structure (via real symmetric matrices) but cannot encode directional information while preserving PSD-ness. In contrast, our complex Hermitian restriction maps preserve the PSD structure (thus enabling Fourier transforms) while allowing directional encoding via their phase. This maintains the essential spectral properties needed to define a convolutional operator without discarding topological information.
>
> Let us note that, ff the PSD/Hermitian structure is not preserved, the spectral interpretation of graph convolutions breaks down. Specifically, a non-PSD matrix can have negative or complex eigenvalues, making the graph Fourier transform ill-defined for filtering purposes: the “frequencies” may no longer correspond to real oscillations on the graph, and spectral multipliers can produce unstable or non-convergent outputs. For example, using a purely real skew-symmetric (antisymmetric) matrix yields purely imaginary eigenvalues, so applying a spectral filter results in oscillatory or diverging behavior rather than meaningful smoothing or directional propagation.
>
> [1] Kipf, T. N. "Semi-supervised classification with graph convolutional networks." arXiv preprint arXiv:1609.02907 (2016).
>
> **Question 1**: Could q be made learnable per edge or per graph?
>
> While the parameter $q$ was treated as a fixed hyperparameter in the main paper, our architecture is compatible with learning $q$ jointly with the model. To address the reviewer’s point, we trained Gen-DSNN with $q$ as a learnable parameter using three different initialization values (0.25, 0.125, 0). Table 1 reports the mean and standard deviation of the learned $q$ over 10 folds for five datasets, together with the fixed $q$ value that was found via hyperparameter selection to lead to the best performance in the original version of the paper paper.
>
> Table 1: Learned q values (mean ± std)
> | Dataset   | init q = 0.25       | init q = 0.125       | init q = 0            | q for best results |
> |-----------|----------------------|------------------------|-------------------------|---------------------|
> | Cora      | 0 ± 0.008            | 0 ± 0.005              | 0 ± 0.004              | 0.10                |
> | Wisconsin | 0.15 ± 0.06          | 0.11 ± 0.03            | 0.05 ± 0.06            | 0.20                |
> | Cornell   | 0.18 ± 0.08          | 0.10 ± 0.09            | 0.01 ± 0.02            | 0.00                |
> | Texas     | 0.15 ± 0.04          | 0.14 ± 0.06            | 0.09 ± 0.08            | 0.10                |
> | Telegram  | 0.19 ± 0.07          | 0.10 ± 0.09            | 0.04 ± 0.05            | 0.20                |
>
> We observe that the learned $q$ converges stably across different initializations: for some datasets (e.g., Cora) it consistently converges to almost zero, suggesting that directional information is less relevant, while for others (e.g., Wisconsin, Texas, Telegram) it converges to moderate values that are close to the best fixed choice used in the original version of the paper. This indicates that learning $q$ is feasible and that the learned $q$ is very much aligned the tuned one. We included this table in Appendix~L and plan to explore a fully learnable or dynamically adaptive $q$ (e.g., per-graph or per-layer scheduling) as future work.

---

> ### Author Response · Authors · 2025-11-20
>
> **Question 2**:  Have you examined the spectral properties or phase distributions of learned complex restriction maps?
>
> We carried out a direct spectral comparison between the Magnetic Laplacian and our Directed Sheaf Laplacian under matched conditions (same values of $q$, stalk dimension ($d=1$), and identical base graphs). This setting allows a meaningful one-to-one comparison, since the Magnetic Laplacian is recovered as the special case of our operator when ($d=$1) and the restriction maps are constrained to be all equal to 1 (for the edges contained in the graph).
>
> Across all tested datasets (Cora, Texas, Cornell) and (q) values, the spectra of the two operators were almost indistinguishable: the eigenvalue distributions aligned extremely closely, and the high-frequency components showed no systematic deviations. This indicates that, in the (d=1) setting, the proposed Directed Sheaf Laplacian preserves the spectral behavior of the Magnetic Laplacian.
>
> We have added a figure illustrating these comparisons in the revised version of the paper (Appendix M).

---

> ### Author Response · Authors · 2025-11-27
>
> Dear Reviewer Xiwg,
>
> Thank you once again for your thoughtful and constructive evaluation of our work. Your questions provided opportunities to clarify and further strengthen the presentation and positioning of the paper. We have made the necessary revisions accordingly.
>
> We believe we addressed all your concerns in our responses. However, we have not yet head from you and would be grateful if you could give us your feedback on our responses, as this would help us ensure that everything has been resolved.
>
> Thank you again for the time and care you invested in reviewing the paper.
>
> The Authors

---

### Author Response · Authors · 2025-12-01
**Summary of Contributions and Rebuttal Revisions**

Dear Area Chair,

Thank you for handling our paper. We are thankful to the reviewers for their positive evaluation of our work (all four provided a score of 6).

Together, they identified three core pillars in our work:

- They found our work "original and mathematically principled", recognizing it as the "the first rigorous sheaf-theoretic framework for directed graphs" capable of "addressing a clear limitation of existing SNNs" for directed graphs. They highlighted how our proposed Directed Sheaf Laplacian provides a "solid theoretical foundation" and a "unifying framework [that] subsumes NSD, MagNet, and SigMaNet as special cases".

- They agreed on our model's effectiveness, noting that DSNN "consistently outperforms GNN and SNN baselines", achieving "top performance on 10/12 node-classification and 8/10 direction-prediction benchmarks, with large margins on heterophilic and directed graphs". They also highlighted its ability to "effectively captur[ing] asymmetric and directional relationships".

- The reviewers emphasized that our work addresses a "clear limitation of existing SNNs" and that the topic of Sheaf Neural Networks applies to directed graphs is "significant". Our "comprehensive experimental results" were seen as "validating the benefits offered by [our proposed model]".

During the rebuttal, we provided an answer to every comment the reviewers made, and added extra results to further solidify our work. In particular:

- We empirically demonstrated that our proposed method (DSNN) scales to large graphs (2.9M nodes), running comfortably on a single 24GB GPU.

- We added four extra recent baselines (DiGCL, HaarNet, CAGN, and H2SGNN), all appeared between 2021 and 2025, to our experiments, demonstrating that our proposed method (DSNN) maintains its performance lead even against the latest state-of-the-art direction-aware models.

- We provided further ablation studies (on learning and tuning q, robustness to noise induced by direction flipping, and stalk dimension d), and added a detailed walkthrough (Appendix J) to make our theoretical contributions accessible to a broader audience.

Taken together, the reviewers’ assessments of the framework’s novelty, theoretical soundness, and empirical performance, along with these additional experiments and clarifications, result in a revised version that we expect to be of genuine interest to the ICLR community.

Best regards,

The Authors

---

### Meta-Review · Area_Chair_Pr4a · 2026-01-07

**Summary:**

This paper introduces the Directed Cellular Sheaf and corresponding Directed Sheaf Laplacian (DSL) and Directed Sheaf Neural Network (DSNN), extending Sheaf Neural Networks (SNNs) to directed graphs through complex-valued restriction maps. The authors provide theorems of spectral properties of DSL. Experimental results on multiple graph datasets are very competitive.

All reviewers agreed to accept this paper, although only one of them possesses high confidence. After checking the rebuttal from authors, I believe most concerns have been addressed.

**Reviewer Concerns:**

All reviewers' comments are almost addressed and discussed during the rebuttal phase.

**Reviewer Scores:**

They may keep their scores.

---

### Decision · Program_Chairs · 2026-01-26

Accept (Poster)